# Seismic events miss important kinematically governed grain scale mechanisms during shear failure of porous rock

Alexis Cartwright-Taylor [1] ✉, Maria-Daphne Mangriotis[1], Ian G. Main [1], Ian B. Butler[1], Florian Fusseis[1], Martin Ling[2], Edward Andò [3], Andrew Curtis [1], Andrew F. Bell [1], Alyssa Crippen[1], Roberto E. Rizzo [1,4], Sina Marti[1], Derek. D. V. Leung [1] & Oxana V. Magdysyuk [5]

Catastrophic failure in brittle, porous materials initiates when smaller-scale fractures localise along an emergent fault zone in a transition from stable crack growth to dynamic rupture. Due to the rapid nature of this critical transition, the precise micro-mechanisms involved are poorly understood and difficult to image directly. Here, we observe these micro-mechanisms directly by controlling the microcracking rate to slow down the transition in a unique rock deformation experiment that combines acoustic monitoring (sound) with contemporaneous in-situ x-ray imaging (vision) of the microstructure. We find seismic amplitude is not always correlated with local imaged strain; large local strain often occurs with small acoustic emissions, and vice versa. Local strain is predominantly aseismic, explained in part by grain/crack rotation along an emergent shear zone, and the shear fracture energy calculated from local dilation and shear strain on the fault is half of that inferred from the bulk deformation.

Catastrophic failure of porous materials is important for a wide range of applications and on a variety of scales, from natural and induced earthquakes to the failure of synthetic materials and engineered structures. Understanding the microstructural processes that operate during the weakening and failure of these materials, and the associated strain partition between seismic and aseismic components is critical to reducing the significant uncertainties involved in inferring local deformation and strain rates from remotely accessed field-scale seismic or geodetic data. Without the ability to reduce uncertainties, we are unable to improve methods for forecasting and mitigating the risks associated with catastrophic failure.

The key driving mechanism of catastrophic failure under triaxial compression is the concentration of precursory damage along localised zones of deformation, eventually resulting in system-sized failure along a distinct and emergent sub-planar discontinuity[1]. Mean field models explain some aspects of the observed behaviour, but rely on average properties that cannot account for localisation or the detailed micro-mechanics[2]. They are often derived solely from the properties of recorded acoustic emissions (AE), which are assumed to be representative of the local strain, and cannot account for local aseismic mechanisms that typically constitute >99%[3] of the total accumulated strain energy. The partition between seismic and aseismic strain directly constrains the rheology and the detectability of seismic precursors to failure and is currently the largest source of uncertainty in the operational forecasting of seismic risk from sub-surface engineering projects[4]. However, we have limited constraints on the strain partition evolution at a large scale, and none at the microscopic scale.

[1]School of GeoSciences, University of Edinburgh, Edinburgh, UK. [2]Independent Electronics Developer, Edinburgh Hacklab, Edinburgh, UK. [3]EPFL Center for Imaging, École Polytechnique Fédérale de Lausanne (EPFL), Lausanne, Switzerland. [4]Department of Earth Sciences, University of Florence, Via La Pira 4, 50121 Florence, Italy. [5]Beamline I12-JEEP, Diamond Light Source Ltd., Harwell Science and Innovation Campus, Didcot, UK. ✉e-mail: alexis.cartwright-taylor@ed.ac.uk

The size, location and fracture mode of individual microcracking events are commonly inferred from AE waveforms, sometimes calibrated against thin sections that reveal the microscopic processes destructively after the test. These processes include elastic compaction, pore collapse, and the development of small tensile microcracks oriented parallel to the maximum principal stress[1,5,6]. Cracking is a form of local failure either due to locally weaker material or local stress concentrations resulting from changes in geometry[7–11]. Under a constant deformation rate, the peak stress often coincides with the onset of a rapid, non-linear acceleration of AE event rate[5,12], resulting in violent, abrupt failure and rapid stress drop. The non-linear rheology and short time scales (<40 s in Clashach sandstone; Supplementary Fig. 3) make the micromechanisms involved in the transition from stable crack nucleation along a localised shear zone to system-sized rupture difficult to capture and characterise in real-time. However, failure can be extended in duration by controlling the loading rate to maintain a constant AE event rate[1,13,14]. This procedure prevents the acceleration of crack damage and can extend to minutes or even hours the microscopic processes that usually occur over a few seconds, enabling the post-peak-stress region to be studied under quasi-static conditions. Mapping AE source locations during such an AE-rate controlled experiment[1,13] provided the first in situ view of microcrack localisation along a shear zone and subsequent shear zone growth by continued microcracking, as well as an estimate of the associated shear fracture energy, a key parameter in the mechanics of earthquakes and faulting. Since then, several in situ acoustic monitoring studies of loading rate effects[15] and AE source mechanisms[16–18] in rocks undergoing deformation have controlled the applied load to maintain a constant AE event rate and slow down the failure process. More recently, in situ high-energy X-ray microtomography (µCT) time-lapse imaging of rock deformation experiments has enabled the non-destructive characterisation of microstructural damage and local strains[19–23]. However, such studies have so far been limited to experiments of constant strain rate or constant stress rate, where changes after peak stress occur too

rapidly to be captured, even with the high-speed imaging capabilities of a synchrotron.

Here, we address this research gap by presenting detailed in situ images of rock failure obtained during experiments in a novel X-ray transparent triaxial deformation cell (see Methods), which for the first time integrates acoustic monitoring (sound) with fast time-lapse synchrotron X-ray imaging (vision), on the beamline I12-JEEP[24] at the Diamond Light Source, UK. We slowed down failure using feedback from the AE event rate[1,14] (see Supplementary Note 1), extended the failure time from ~1 to ~50 min (Supplementary Fig. 3), and captured sample weakening and failure in a series of eighteen 3D X-ray µCT volumes (Fig. 1a), along with ultrasonic velocity changes and the locations and amplitudes of the AE sources. This allowed us to unlock the relationship between the seismic sources, the local strain and the associated underlying micro-mechanisms. The results provide an integrated picture of how damage and associated micro-seismic events emerge and evolve together during localisation and failure and allow us to ground truth previous inferences from mechanical and seismic monitoring alone. We find that such inferences miss important developments on the grain scale prior to and during failure; early strain localisation does not necessarily lock the system in; and the micro-mechanics on the grain scale are governed by kinematics, in turn, governed by the rock configuration. This improvement in process-based understanding offers the prospect of reducing systematic errors in forecasting system-sized catastrophic failure in a variety of applications.

## Results

### Micro-mechanics of strain localisation and failure from in situ X-ray images

Under AE-controlled quasi-static loading, the sample eventually experienced system-sized brittle failure along a localised shear zone (Fig. 1), with X-ray µCT volumes of the precursory microscale processes obtained at the times indicated. The details of damage localisation and shear zone development are shown in post-yield 2D µCT slices (Fig. 2) and incremental 3D strain fields obtained by digital volume correlation (Fig. 3).

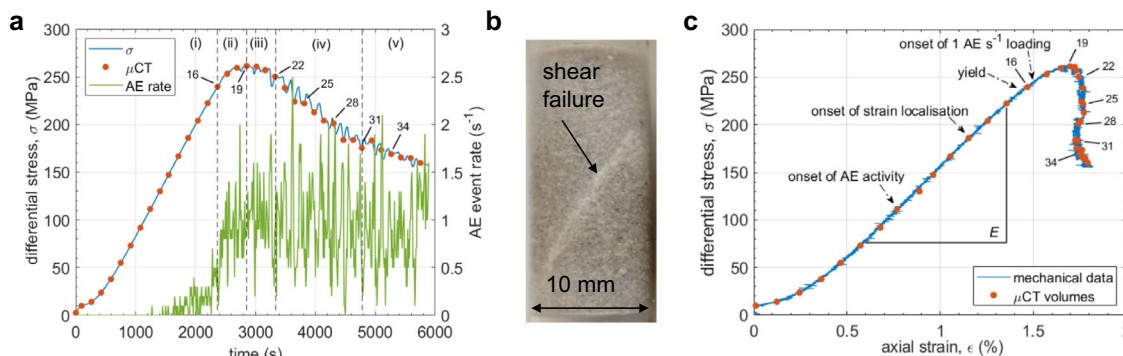

**Fig. 1 | Bulk mechanical behaviour of Clashach sandstone. a** Evolution of differential stress, $\sigma$ (blue line), and acoustic emission (AE) event rate, $\dot{N}$ (green line), with time. Orange circles denote the times of the X-ray tomographic volumes. The plotted AE event rate was calculated from all recorded events, binned into 10 s time intervals. Consistent with previous studies[75,76], we identified five stages of deformation: (i) initial compaction and then quasi-elastic behaviour up to the yield point, (ii) strain hardening approaching peak stress, $\sigma_P$, (iii) damage zone localisation and strain softening beyond $\sigma_P$, (iv) sample weakening due to shear zone development through the sample and (v) shear sliding along a contiguous sub-planar fault. The transition from constant strain rate loading ($10^{-5} \, s^{-1}$) to constant AE event rate loading ($1 \pm 1 \, AE \, s^{-1}$) occurred early in stage (ii) shortly after the sample yield point, which was defined by the point at which the stress-strain curve deviated from linearity and the AE event rate accelerated beyond the steady but low rate observed during the elastic region (linear portion of the stress-strain curve). **b** Photograph of

the failed sample showing the localised shear damage zone. **c** Differential stress plotted against axial strain. Number labels in **a** and **c** refer to the tomogram slice and strain increment labels in Figs. 2 and 3, with tomogram 16 acquired at the yield point (transition from stage i–ii), tomogram 19 acquired at peak stress (transition from stage ii–iii), tomogram 22 acquired as microcracks localised along the critically oriented shear zone (transition from stage iii–iv), tomograms 25 and 28 acquired during shear zone development, tomogram 31 acquired at the onset of coherent sliding (transition from stage iv–v) and tomogram 34 acquired during coherent sliding. Young's modulus, $E = 19.369 \pm 0.028$ GPa, was calculated over the range shown. AE activity began at 40% of peak stress, $\sigma_P$, with initial strain localisation evident in the strain increments from $0.7\sigma_P$ onwards, and sample yield following at $0.85\sigma_P$. The AE feedback control ($1 \, AE \, s^{-1}$) modulated the strain rate from $0.93\sigma_P$.

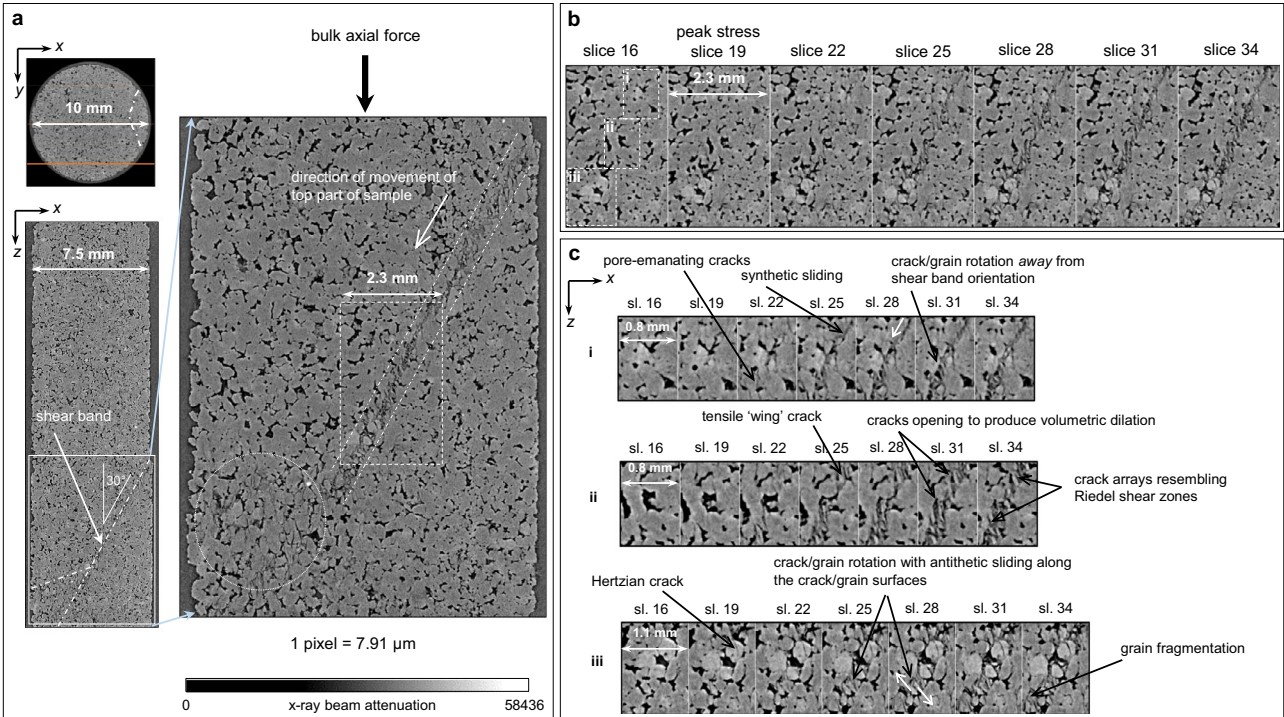

**Fig. 2 | Microscale damage evolution close to and following peak stress.**
**a** Reconstructed 2D X-ray microtomographic (μCT) slice (*x*,*y*-oriented, where *x* and *y* are perpendicular to each other and to the direction of loading, and *x* is across-strike and *y* is along-strike of the shear zone) showing the plane (orange line) of a re-slice of the original μCT volume (*x*,*z*-oriented, where *z* is the direction of loading) where the shear zone initially localised. The corresponding across-strike (*x*,*z*-oriented) re-slice is shown directly below the *x*,*y*-oriented slice with the shear zone highlighted by dash-dot lines. Zoomed-in view of the portion of the *x*,*z*-oriented re-slice contained within the solid box is shown between the pale blue arrows, high-lighting the narrow shear zone that formed after peak stress (between the dash-dot lines), and the region of damage that formed after yield but before peak stress (dotted circle). **b** Further zoomed-in view of the across-strike slices (*x*,*z*-oriented) for selected tomograms, labelled with the number by which they appeared in the time-series (see Fig. 1 for the locations of these scans in the stress-time and stress-strain evolution, and Fig. 3 for the local strain increments following each of these scans) showing shear zone emergence and development in region shown by dashed box in zoomed-in view in **a**. **c** Even further zoomed-in slices (*x*,*z*-oriented) high-lighting the variety of micro-mechanisms involved in shear zone formation: num-bers i–iii correspond to the dashed boxes in (**b**; slice 16). When the whole time-series is viewed as an animation (Supplementary Movies 1 and 2), the micro-mechanisms illustrated by the annotations are apparent. The grey-scale intensity represents the X-ray beam attenuation to the rock. Darker pixels represent struc-tures with less beam attenuation, i.e. lower density structures such as pores or cracks. Lighter pixels represent structures with more beam attenuation, i.e. higher density structures such as mineral grains.

Initially, diffuse elastic compaction was observed throughout the microstructure (Supplementary Figs. 6b and 17). AE activity preceded initial localisation of dilation and shear strain during early loading [stage (i) in Fig. 1; Fig. 3 panel 13], consistent with previous AE studies on porous rocks using much larger samples[5,6,25] or in situ μCT imaging of smaller samples[22,23]. Between yield and shortly after peak stress [stage (ii); Fig. 3 panels 16–19], the spatial distribution of shear strain closely followed that of dilation, and competing strain clusters loca-lised along three distinct conjugate planes of similar amplitude and dip (30° to maximum principal stress; typical of optimally-oriented faults in nature) but variable strike, indicating self-organised exploration of candidate shear zones. These direct observations highlight the exploratory nature of emergent localisation in a complex system. Microcrack damage initiated towards the bottom end of the sample (Fig. 2a), where a region of localised compaction, evident from enhanced vertical displacement (Supplementary Fig. 15a), led to the collapse of some pores and facilitated the subsequent nucleation of pore-emanating microcracks. Some of these cracks traced grain boundaries (inter-granular cracks), while others intruded into whole grains (intra-granular and trans-granular cracks). Some trans-granular cracks initiated at loaded grain–grain contacts, most likely due to local Hertzian contact forces within larger-scale force chains of accumu-lated stresses. These cracks formed subparallel to the loading axis, were no longer than two grain diameters, and were observed to cluster towards the bottom end of the sample (Fig. 2a), in the region of

competing strain clusters (Fig. 3; panels 16–19) approaching peak stress. The enhanced vertical displacement in this part of the micro-structure facilitated the bulk shear movement of the top part of the sample towards the weakened region of compacting porosity and tensile microcracking. This, in turn, led to further strain localisation along the candidate shear zone that was critically oriented for failure, and coherent relative movement of the 'hanging wall' above the shear zone (Supplementary Fig. 15b).

In stage (iii), dilation and shear strain concentrated along the critically oriented shear zone soon after peak stress (Fig. 3 panel 22), preceded by a brief hiatus in the dilation and shear strain rate (Sup-plementary Fig. 4; panel 20). This hiatus was consistent with a similar hiatus observed in AE event rate shortly before failure[26], and in our case, was associated with a brief increase in the rate of diffuse com-paction (Supplementary Figs. 16 and 17; panel 20). The critically oriented shear damage zone emerged spontaneously from the self-organised localisation of numerous, narrow en-echelon tensile microcracks that nucleated simultaneously along the whole length of the emerging shear zone (Fig. 2b and c; slice 22) due to localised, high amplitude dilation and shear strain (pink and green regions respec-tively in Fig. 3; panel 22 and Supplementary Fig. 4; panels 21 and 22). These en-echelon microcracks were, individually, predominantly confined to single whole grains and originated from pores and Hert-zian contacts. As tensile damage mechanisms localised increasingly on the shear zone, initial diffuse compaction throughout the sample was

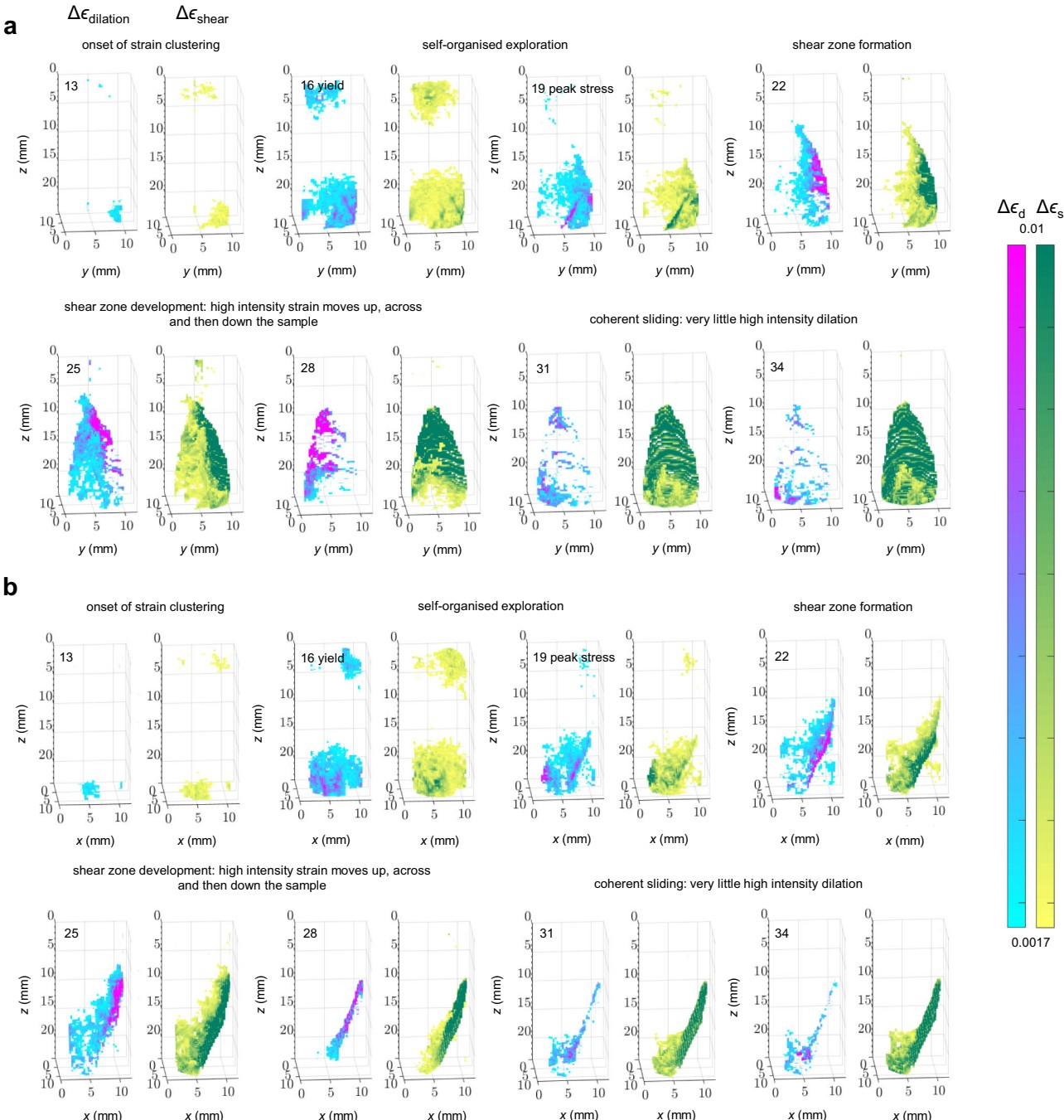

**Fig. 3 | Selected 3D incremental strain fields from the onset of strain clustering (marked in Fig. 1c).** Incremental dilation, $\Delta\epsilon_d$ (blue-pink), and shear strain, $\Delta\epsilon_s$ (yellow-green), were calculated from digital volume correlation between successive pairs of tomographic volumes and are shown **a** parallel to strike ($y,z$ orientation) and **b** perpendicular to strike ($x,z$ orientation). The lower threshold of 0.0017 was set at four standard deviations from the mean of the error distribution of $\Delta\epsilon_s$ (Supplementary Fig. 6) and the upper threshold shows regions with strain >0.01 (maximum $\Delta\epsilon_s$ and $\Delta\epsilon_d$ were -0.04; Supplementary Figs. 4 and 5). The thresholds were chosen to visually highlight regions of localised strain. Number labels correspond to those in Figs. 1 and 2, with the strain increment between the numbered tomogram and its subsequent neighbouring tomogram. Strain localisation began at 70% of the peak stress, $\sigma_P$. The yield point here is the

same as that shown in Fig. 1c, with the corresponding strain increment shown immediately following yield. The shear zone formed as crack localisation occurred along the critically oriented plane in tomogram 22 (Fig. 2), and developed between strain increments 22 and 31, with patches of high intensity strain moving first up, then across and eventually down the sample, until high intensity dilation almost stopped. We defined coherent sliding as sliding along the whole shear zone, with the onset of coherent sliding at strain increment 31 since the intensity of dilation was significantly less in this increment than in previous increments, and this increment coincided with the point at which the rate of stress reduction slowed down in Fig. 1a. The full time-series of incremental dilation and shear strain from the onset of strain localisation is shown in Supplementary Fig. 4.

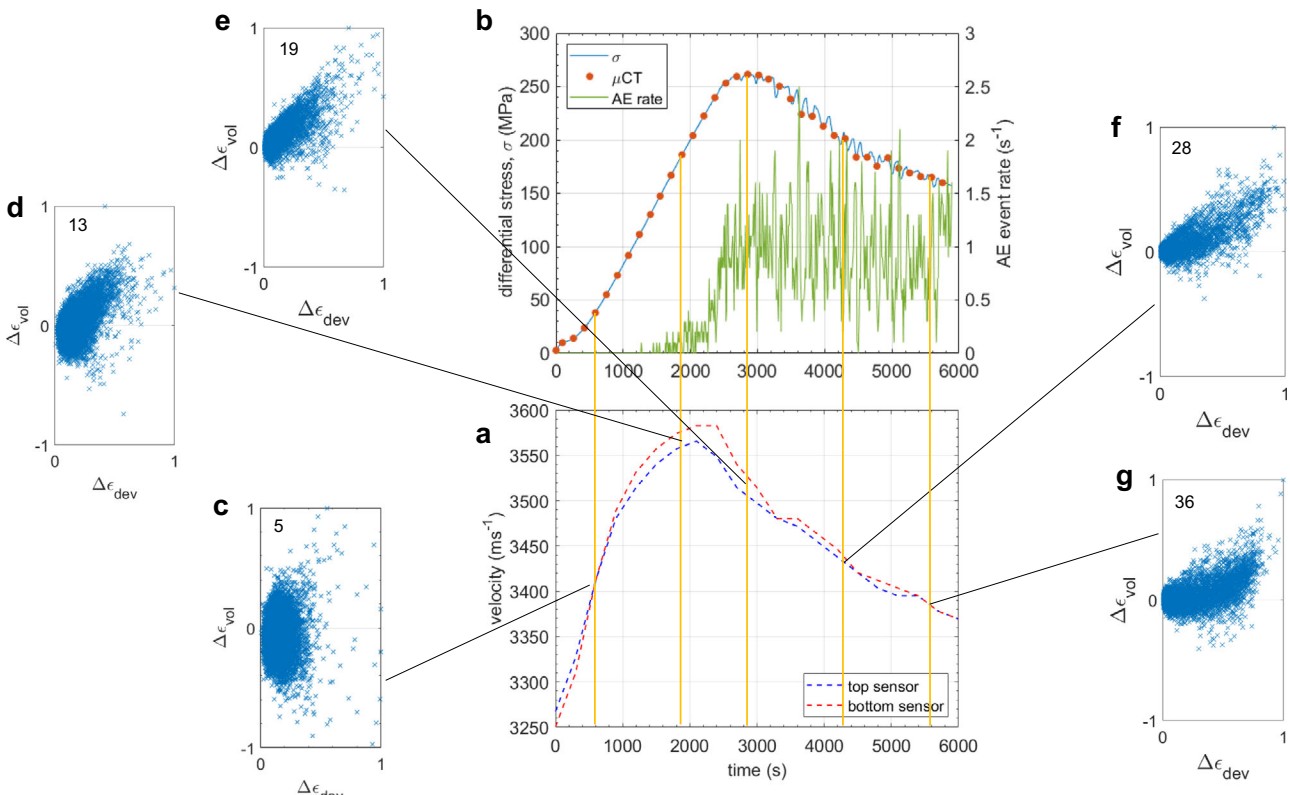

**Fig. 4 | Velocity, stress and AE event rate evolution as a function of time, with selected incremental strain field cross-plots. a** Velocity evolution, defined from active seismic surveys, using the top sensor as a receiver with the bottom sensor as the seismic source (blue line) vs. the bottom sensor as a receiver with the top sensor as the seismic source (red line). **b** Differential stress throughout loading (blue line) along with AE event rate (green line). Orange circles denote the times of the X-ray tomographic volumes. **c–g** Normalised cross-plots of the local incremental deviatoric (shear) strain, $\triangle\epsilon_{dev}$, vs. incremental volumetric strain, $\triangle\epsilon_{vol}$, showing the initial transition from **c** compaction-dominated strain to **d** dilation-dominated strain, **e** the correlation between dilation and shear strain through localisation and shear zone development, **f** the relaxation of dilation once the shear band was fully developed and finally **g** shear-enhanced compaction during coherent sliding. Dilation is defined as positive volumetric strain and number labels refer to the position of the strain increment in the time-series (as in Fig. 1a). The full time-series of cross-plots is shown in Supplementary Fig. 16.

swamped by localised dilation and shearing on the shear zone (Supplementary Figs. 4, 6, 16 and 17; Supplementary Movies 3–8). This was marked by the emergence of a fat tail in the respective frequency-amplitude distributions, which eventually became bimodal (Supplementary Fig. 5).

In stage (iv), the shear zone developed along-strike (Figs. 3 and 4; Supplementary Fig. 12). It developed with a degree of curvature, consistent with shear zone development on a crescent-shaped front revealed by AE locations in granite[21]. This along-strike development, together with the observed variation in strain intensity within the shear zone, is contrary to the assumptions of the breakdown zone model, which assumes a uniform slip distribution across a sample-sized fault (see Methods). Dilation and shear strain were highly correlated in the shear zone (Fig. 3), consistent with several micro-mechanisms co-existing to accommodate bulk shear motion (Fig. 2c; slices 22–31). These included the nucleation of pore-emanating and Hertzian en-echelon, tensile cracks along the shear zone, which facilitated further downslope bulk shear movement of the top part of the sample. The opening of these cracks caused some of the new tensile cracks to widen, producing dilation and new pore space and promoting synthetic sliding (parallel to the principal slip direction) on cracks favourably oriented parallel to the shear zone. This, in turn, led to the development of tensile wing cracks at the tips of these sliding shear cracks. These micro-mechanisms are consistent with previous experimental observations and existing microcrack nucleation models[7–11]. Bulk shear motion along the failure plane also caused some en-echelon tensile cracks to rotate away from the shear

zone orientation. These were cracks that had dilated sufficiently to allow neighbouring grain fragments to rotate with the bulk shear motion of the top part of the sample. Some of these grain fragments remained attached as asperities to the walls of the shear zone, while others broke away with continued bulk shear motion. Rotation prevented further tensile crack propagation in the axial direction and supported the walls of the shear zone to maintain a finite thickness of up to one grain diameter throughout failure, although a few individual cracks extended up to two grain diameters. It also facilitated antithetic motion (conjugate to the principal slip direction) along cracks oriented conjugate to the principal shear zone (unfavourably oriented for principal slip), including some resembling Riedel shear zones. Some rotating fragments moved freely in the newly generated pore space, antithetically relative to their neighbouring grains but without contact between them. We would expect these movements to be aseismic. Other fragments were close enough for their crack surfaces to remain in contact during antithetic sliding against each other. We would expect these movements to be seismic. Local crack/grain rotation with associated antithetic motion occurred frequently along the length of the shear zone and was apparent in every vertical slice along the strike direction. The grey-scale μCT images show that this mechanism was most prevalent within the shear zone on the side of the sample where microcrack localisation was initiated along the emerging failure plane. Further along the strike, parts of the shear zone became narrower (less than one grain diameter), with vertical displacement increasingly accommodated along more steeply dipping and narrow

tensile and shear fractures via a wing-crack style mechanism. Rotation was still apparent in less steeply dipping regions, although there was an overall decrease in the number and size of rotating grains/cracks, and in the total angle rotated, further along the strike. All the mechanisms just described led to grain fragmentation (cataclasis), and generated a proto-cataclasite within the shear zone as whole grains disintegrated, partial grains fractured off the shear zone walls, and fractured grains filled cavities (Fig. 2c; slices 25–34). Ongoing cataclasis resulted in compaction along the shear zone, spatially correlated with dilation and shear strain, but with smaller amplitude (Supplementary Figs. 4 and 17; Supplementary Movies 3–8). In addition to the compaction of new pore space that had been generated by dilation within the shear zone, grain/crack rotation was a key facilitator of shear zone compaction. Increased rotation caused the crack dip angles to decrease, discouraging continued shear motion between grain fragments while encouraging compaction and closure of shallow dipping cracks during coherent sliding. These observations highlight the significant aseismic contribution (i.e. rotation of freely moving grain fragments, and other silent grain rearrangements) to the overall failure process, relative to seismic mechanisms (i.e. cracking and dynamic shear sliding[6]).

Eventually, coherent slip occurred on a contiguous fault plane that reached the sample boundary on all sides (stage v; Fig. 3 panels 31–34), with the corresponding reduction of local dilation and shear strain amplitude and local strain rate. This occurred as the main failure mechanism transferred from local breakdown and rotation to coherent slip with a component of shear-enhanced compaction (Supplementary Figs. 4, 5, 16 and 17). All the micro-mechanisms just described remained active in accommodating coherent slip, explaining the remaining patches of dilation (Fig. 3 panels 31–34), including rotation of fragments still attached to the walls of the shear zone forming local asperities. As these asperities broke away, synthetic shear cracks/voids developed along the walls of the shear zone, facilitating further sliding.

In summary, we observed a breakdown sequence from failure of individual grains due to point loading, deformation of pores and local shear sliding, through the formation of a proto-cataclasite due to grain rotation and fragmentation, to an ultra-cataclasite due to further grain fragmentation, granular flow and shear-enhanced compaction during coherent slip.

## Seismic signature: velocity, AE and the seismic strain partition

Ultrasonic velocity surveys, with source-receiver geometries at opposite ends of the sample (Supplementary Fig. 1), were performed every five minutes throughout the experiment to characterise the compressional-wave velocity, $V_P$, along the loading direction, and to locate the acoustic emissions (AE). Figure 4 shows that $V_P$ initially increased in response to compaction (aforementioned stage i) and then decreased during strain hardening (stage ii), in line with the observed transition from compaction-dominated to dilation-dominated local strain at the yield point, and concurrently with exploratory strain localisation (Fig. 3). Beyond peak stress, strain softening (stage iii) and shear zone propagation (stage iv) were marked by a continued decrease in $V_P$ due to dilatant microcracking and newly generated pore space along the shear zone associated with localised dilation and shear strain (Figs. 2, 3 and 4). $V_P$ continued to decrease throughout coherent slip (stage v; tomograms 31-end), although at a slightly reduced rate, indicating continued but reduced dilation as the local deformation mechanism became predominantly shear with an increased contribution from compaction (Fig. 4), consistent with previous observations of AE source types[6,16–18]. However, $V_P$ never recovered to its original value, indicating greater early compaction than subsequent dilation along the direct arrival path of the axial P-wave.

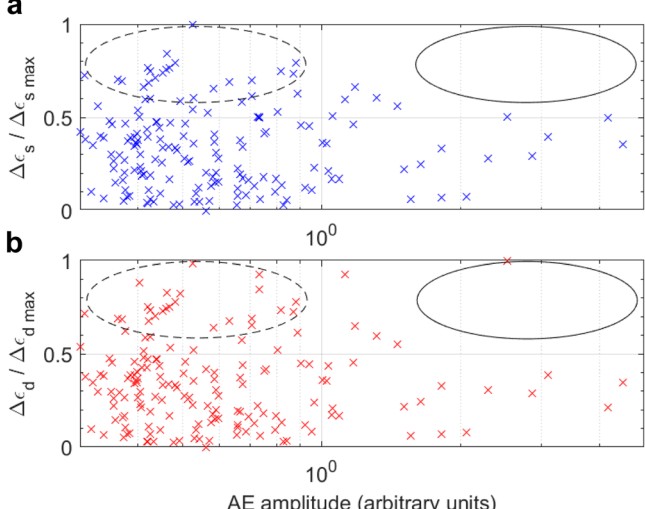

**Fig. 5 | Relationship between acoustic emission (AE) amplitude (in log-scale) and local incremental strain at the AE location. a** Incremental shear strain, $\Delta\epsilon_s$ (blue crosses), and **b** incremental dilation, $\Delta\epsilon_d$ (red crosses), each normalised by their respective maximum. The solid black ellipses highlight the lack of large AE events at locations of large strain; indicative of predominantly aseismic deformation. Conversely, the dashed black ellipses highlight locations of large strain which are connected to mostly smaller AE events. Larger AE events are mostly associated with smaller strains.

We recorded ~3600 AE events above ambient noise (instantaneous amplitude threshold of 280 mV at 70 dB pre-amplification gain) using axially located P-wave sensors (see Methods). Some 5% of these events were selected using objective criteria for location analysis (see Methods). The most likely location for each event was inferred by constraining it to the maximum local strain within a kinematically-derived hyperboloid of possible locations (see Methods and Supplementary Fig. 11). The same unique location was found for 85% of the located AE events in both the dilation and shear strain fields due to the high correlation between the two strain fields. Even with the maximum strain constraint, the largest AE events did not occur at locations of high local strain (Fig. 5; solid black ellipses), implying the seismic strain partition coefficient is highly variable in space. Many small events occurred in regions of high local strain (Fig. 5; dashed black ellipses), indicative of deformation being primarily aseismic in the shear zone, while many large events occurred in regions of low local strain in bulk (Fig. 5 and Supplementary Fig. 12). Dilation and shear strain are positively correlated over the whole experiment, but this positive correlation is much stronger at the AE locations (Supplementary Fig. 13), consistent with a significant proportion of mixed-mode (significant shear and tensile components) seismic moment tensors observed in earlier studies[6,18].

We inferred a seismic strain partition coefficient for bulk deformation of 0.5% from summing the inferred scalar seismic moments (see Methods), verifying that deformation is primarily aseismic. This is much lower than the 1% inferred by Dresen et al.[3], most likely due to using a less brittle material (sandstone) under quasi-static rather than constant strain rate loading. Both estimates are a lower bound due to the finite signal-to-noise ratio.

## Rupture energy

The shear fracture energy or energy release rate, $G_c$, is a crucial parameter for modelling shear fracture propagation[27]. It is the energy required per unit area for breakdown processes, such as tensile fracturing, to create the new fault surface. It characterises the strain-softening region of the stress-strain curve and is conventionally estimated from bulk axial stress and strain data (Fig. 6a). Here, we estimate $G_c$ both from bulk and from knowledge of the local strain in the shear

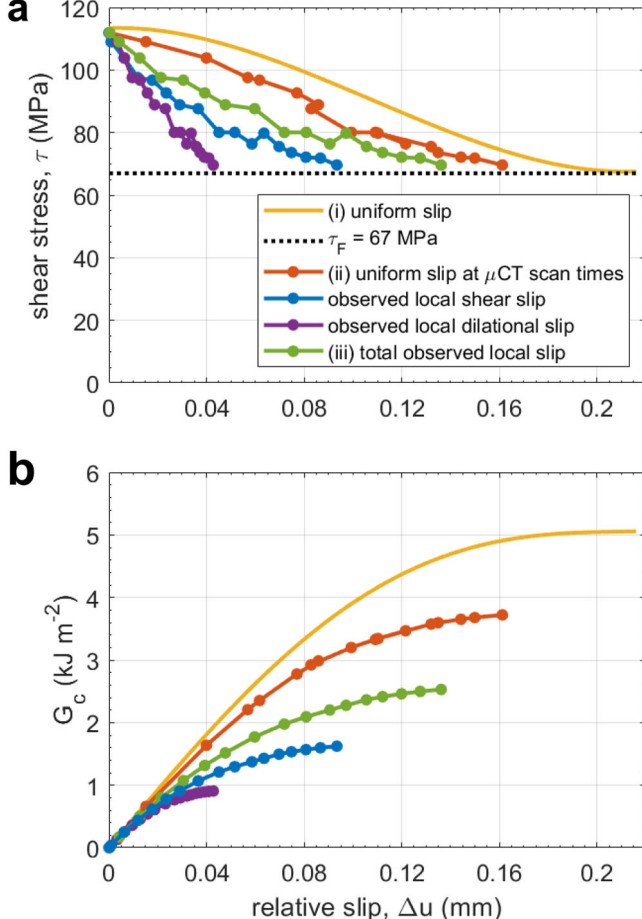

**Fig. 6 | Shear fracture energy estimated from bulk strain and from local incremental strain in the shear zone. a** Relationship between the shear stress acting on the shear zone, $\tau$ and relative slip, $\Delta u$ for average angle $\theta = 30.3°$ that the shear zone makes with the axial differential stress, $\sigma_1$ (see also Supplementary Table 1). We show relationships for (i) uniform slip, estimated according the method described in Wong et al.[28] (see also Fig. 1. in Wong et al.[28]), from average bulk axial stress and strain data (yellow), (ii) uniform slip, again estimated as per Wong et al.[28], from actual bulk axial stress and strain measurements at each $\mu$CT scan time (orange) and (iii) total observed local slip in the developing shear zone (green), obtained from direct dilation and shear strain measurements (see Methods). We obtained the shear fracture energy, $G_c$ by integration under the $\tau$-$\Delta u$ curve from peak shear stress, $\tau_P$ to the residual frictional strength $\tau_F$ (dotted black line) of the shear zone. At $\tau_F$, the shear zone reached the critical slip distance, $\Delta u^*$ at which breakdown transitions to frictional sliding. The contributions to (iii) from local shear strain (blue) and local dilation (purple) are also shown. **b** Evolution of $G_c$, calculated from **a**, as $\Delta u$ increases.

zone (see Methods), finding that bulk estimates are biased to large values, and the local critical slip distance is smaller than predicted by the bulk model.

We computed $G_c$ (Fig. 6c) by integrating the shear stress versus fault slip record (Fig. 6b) for (i) uniform slip on a sample-sized fault from bulk axial stress and strain[28], (ii) uniform slip from bulk axial stress and strain at each $\mu$CT scan time, and (iii) total observed local slip measured directly from local dilation and shear strains within the shear zone (see Methods for details). We observed a smaller critical slip distance, $\Delta u^*$ (Fig. 6a) for local slip (0.14 mm) than for uniform slip (0.165–0.2 mm), while the integrated curves yielded $G_{c-i}$ = 5.06 kJm$^{-2}$, $G_{c-ii}$ = 3.72 kJm$^{-2}$ and $G_{c-iii}$ = 2.53 kJm$^{-2}$, where dilation and shear strains contributed 36% and 64% of $G_{c-iii}$, respectively. The discrepancy between $G_{c-i}$ and $G_{c-ii}$ arises from the feedback control: many of the $\mu$CT scan times after peak stress coincided with a reduction in stress

and strain as the ram backed off to maintain the AE event rate (Fig. 1a). Nevertheless, local shear fracture energy, $G_{c-iii}$, is only 0.68$G_{c-ii}$ and 0.5$G_{c-i}$. Hence, bulk estimates of $G_c$ for uniform slip across a sample-sized fault can significantly exceed those determined directly from local slip measurements in the developing shear zone.

## Discussion

Our experiment provides an integrated view of crack localisation and shear zone development, combining both in situ X-ray $\mu$CT and in situ acoustic data. It validates many inferences from classic AE experiments, such as the nucleation and growth of a shear zone containing the eventual fault plane due to the spontaneous localisation of en-echelon tensile (dilatant) microcracks[1,13]. It also demonstrates that shear and compactant micro-mechanisms become increasingly important during shear zone development[6,16–18]. Several of the observed micro-mechanisms of deformation, such as tensile micro-cracking within and between grains by pore-emanating and Hertzian contact mechanisms, and the strong correlation between dilation and shear strain, are consistent with those observed in previous in situ $\mu$CT studies up to peak stress[19–23]. Our data also provide insight into the micro-mechanics of strain localisation and shear failure, notably:

- the grain size control on failure;
- the self-organised exploration of candidate shear zones close to peak stress;
- the continued correlation between dilation and shear strain throughout sample weakening;
- the relative importance of aseismic mechanisms such as crack rotation in accommodating bulk shear deformation;
- the lack of correlation between locations of large seismic events and regions of high local strain;
- the very low and locally highly variable seismic strain partition coefficient; and
- the relatively low local shear fracture energy compared to bulk estimates.

The axial $V_P$ evolution is in agreement with published laboratory measurements[13,15–18]. The initial increase in velocity occurs in response to pore size reduction and closure of intra- and inter-granular cracks and high aspect-ratio pore spaces, which leads to a reduction in excess compliance[29]. In response to a competing mechanism, dilatant microcracking, which tends to reduce the velocity, there is a non-linear reduction in the rate of $V_P$ increase until dilatant microcracking dominates at the yield point and the velocity shows an overall reduction thereafter. In our case, $V_P$ never recovered to its original value, consistent with earlier observations[16–18]. This reflects the increasingly heterogeneous damage evolution, whereby axial $V_P$ is less sensitive to the mainly radial dilation that occurs during tensile microcracking, consistent with observations of increasing ultrasonic velocity anisotropy in larger samples[15–18]. The difference from earlier studies is that we have independently verified the underlying mechanisms of compaction and dilatancy from the strain fields, hence validating hypotheses derived from these earlier studies.

Our combined direct ($\mu$CT) and indirect (AE) in situ observations show a tendency for the AE sources to be initially more broadly distributed throughout the sample (strain increments 1–19) and then to progressively localise along the three candidate shear zones close to peak stress and, eventually, the critically oriented shear zone (strain increments 19 onwards). Although we were limited by the number of AE events (<200 events, representing ~5% of recorded AEs) we could locate with our location algorithm, this overall trend is consistent with previous in situ observations of AE localisation in larger samples[1,13,15–18]. Polarity estimates from our AE dataset were unreliable and insufficient, preventing us from distinguishing AE source types and testing hypotheses regarding the relationships between AE amplitude and source type or between AE source type and local strain magnitude.

However, the strain field evolution (Figs. 3 and 4 and Supplementary Figs. 16 and 17) is broadly consistent with earlier observations of AE source types during shear failure[6,17,18], which show a high proportion of tensile-type events approaching peak stress, decreasing during failure in favour of an increasing proportion of shear- and collapse-type events. Furthermore, the observed co-existence of tensile, shear and collapse micro-mechanisms within the shear zone, along with tensile microcracks emerging from pore collapse, Hertzian grain contacts and shear sliding, reflects the relatively high proportion of mixed-mode AE sources previously detected during shear failure[6,18].

The Gutenberg-Richter $b$-value estimate from the AE was $1.94 \pm 0.04$ (Supplementary Fig. 10), slightly higher than $b$-1.7 estimated for granite under similar loading conditions[1]. Both are likely high due to the forced quasi-static failure protocol, and our estimate may also be biased further to high values by the narrow dynamic range of the measurements[30]. When $b > 1.5$, the moment release is dominated by smaller events, so our very low estimate for the seismic strain partition factor (0.5%) may be due in part to the lack of detection of smaller events above the relatively high ambient noise present in the synchrotron beamline. However, this does not explain the relative absence of large AE events in high-strain regions (Fig. 5); a behaviour that would not change significantly if we included the unlocated 95% of events since those events were smaller than the located events. The distribution of AE locations reflects the high $b$-value, with many moderate/small events occurring in regions of large directly measured strain. One explanation could be that the reduction in local stiffness with increasing damage leads to the preference for smaller events along the shear zone, with fewer but larger AE occurring at locally stiff regions off-fault where strain energy accumulates with relatively little deformation. These results show that the seismic strain partition coefficient is highly variable locally, and therefore the AE source amplitude is not necessarily representative of the local strain. This is consistent with similar counterintuitive scaling shown between avalanche size (related to strain) and average energy (related to AE amplitude) in a local load-sharing fibre bundle model[31]. These findings may help to explain similar spatial variability inferred at the field-scale and constrain model uncertainty when forecasting seismic risk[4].

While the DVC correlation window size (316 μm) of approximately one grain size (250–400 μm) averaged over processes occurring at sub-grain size, DVC estimates of the displacement of these windows were accurate to sub-voxel (>7.91 μm) resolution[32–34]. We, therefore, expect the local strain values to be accurate and representative of sub-grain-scale deformation, but unable to discriminate between sub-grain-size micro-mechanisms. One reason for the observed lack of correlation between large AE events and large strains may be a combined spatio-temporal resolution constraint, whereby instantaneous AE sources of different types, occurring very close together over a time-frame shorter than the inter-scan time (~85 s), may cancel each other out. For example, this would suppress local strain estimates in areas where pore collapse also initiated tensile pore-emanating cracks, within the shear zone where radial dilation co-occurred with axial compaction, or where synthetic shear sliding co-occurred with anti-thetic shear sliding. This would lead to instantaneous AE events being correlated with smaller strains than perhaps they should have been. All of these could potentially alter the strain–AE amplitude mapping shown in Fig. 5.

Locations of dilation were strongly correlated with those of shear strain throughout the processes of localisation under the conditions examined here ($P_{eff} = 20$ MPa). This applied equally to the exploration of candidate shear zones and the ultimate development of the final shear zone. Although localised compaction along the critically oriented shear zone did occur (Supplementary Fig. 17; panel 24) this was significantly lower in intensity prior to the coherent sliding phase. Thus, shear zone development was primarily enabled by localised dilation. This is consistent with the independent observation of

dilatant shear zones in post-failure μCT images of dry Vosges sandstone and post-failure microscopy images of saturated Berea sandstone[35,36] after deformation under a constant strain rate at pressures within the brittle regime ($P_{eff} < 40$ MPa). However, it is at odds with observations of mainly compactant shear zone development during the deformation of Flechtingen sandstone[17,18] at $P_{eff}$ of 40 MPa under a constant AE event rate. These differences may be explained by inferences that the transition to shear-enhanced compaction occurs at the transition from the brittle regime to the semi-brittle regime[35,36] ($P_{eff}$ ~ 40 MPa), and could indicate that effective pressure conditions have more influence on the micro-mechanics than differences in loading rate.

The strong correlation between local dilation and shear strain (Fig. 3 and Supplementary Figs. 4, 6 and 16; panels 13–19) is consistent with in situ μCT observations up to peak stress[20]. Here, we show that this correlation continued throughout quasi-static failure, confirming, at higher resolution, inferences from seismic velocity tomography[37]. If volumetric dilation is a proxy for tensile cracking, the correlation confirms the existence of a cohesive zone, but with crack damage distributed throughout the shear zone rather than concentrated solely in a breakdown zone at the propagating front of a pre-existing discontinuity, as proposed by Barenblatt[38]. This observation is consistent with the fact that no contiguous fault exists until very late in the process, and even then, cracking continues to occur at the grain scale by fracturing individual grains and breaking remaining grain-scale asperities distributed across the whole shear zone. However, en-echelon tensile cracks are the first of the damage micro-mechanisms to occur as the shear zone moves across the sample, with the more aseismic processes of further dilation, rotation and cataclasis following behind. This two-stage fault weakening process is independently consistent with the slip-weakening curve for the total observed local slip in the developing shear zone (Fig. 6b; green): a short, steep reduction in shear stress is followed by a longer, shallower reduction. Our observations of two-stage weakening are consistent with observations of near-tip weakening followed by long-tailed weakening in biaxial stick-slip experiments[39]. En-echelon tensile microcracking was not observed along the shear zone until tomogram 22 (i.e. once dilation intensities in Fig. 3a reach into the purple values), indicating a critical amount of dilation is required for microcrack localisation along the shear zone. Therefore, the region of low amplitude strain that precedes higher amplitude strain as the shear zone grows across the sample (Fig. 3a and Supplementary Fig. 4a) is likely not a breakdown zone in the micro-mechanical sense.

Our estimate of $G_{c-i}$ (Fig. 6c) is twice that estimated for Berea sandstone under the same assumptions[13], and an order of magnitude smaller than reported for granite[1,28,37]. Since $G_{c-iii}$ for the local slip in a propagating shear zone is only 50–68% of our two $G_c$ estimates (i and ii, respectively) for uniform slip on a sample-sized fault, it is possible that significant slip distributed throughout the rest of the sample, including in the candidate localised shear zones, may account for the discrepancy between the local and bulk shear fracture energies (~32–50%). However, the average off-fault to on-fault incremental shear and volumetric strain ratios during failure are only 9% and 3%, respectively (Supplementary Table 1); evidence for only a small amount of distributed slip, and consistent with ratios of off-fault dissipated energy to on-fault shear fracture energy in granite[37]. This means that, although not all axial strain was accommodated by the shear slip in the fault plane (contrary to one of the assumptions of the breakdown zone model—see Methods) and therefore the estimated bulk shear fracture energy for the eventual fault plane is an upper bound, slip in the shear zone still dominated the total. Therefore, the remaining 20–38% of additional bulk shear fracture energy must be accommodated by slip within the shear zone itself that does not occur by dilation or shear mechanisms. Such mechanisms are more likely to be aseismic

(i.e. crack and grain rotation and other silent grain rearrangements, including antithetic relative motion of non-touching grains and grain fragments), rather than seismic (i.e. dilation induced by tensile microcracking, and dynamic shear sliding along narrow, rough crack or grain boundary surfaces that are in contact with each other or have asperities between them). While more detailed quantification of the contribution of rotation to the shear fracture energy is required, slip due to rotation for individual cracks can be as large as $77 \pm 29\%$ of the local relative slip (Supplementary Table 2), and hence likely accounts for a significant proportion of the shear fracture energy.

Our results highlight the complexity of damage processes occurring within shear zones during localisation and through weakening and failure. The quasi-static nature of the shear zone development presented here suggests that our results may be most directly applicable to slow earthquakes, but it is also possible that all the observed processes also occur during dynamic failure, just much more rapidly and potentially all together. Fault propagation rates independently inferred from high-resolution AE records[15] were three orders of magnitude smaller during AE feedback loading than during constant strain rate loading ($3–14 \mu s^{-1}$ and $1–18 \, mm \, s^{-1}$, respectively). However, during constant strain rate loading, fault propagation underwent a stable growth phase between initial fault localisation and unstable dynamic propagation, implying mechanistic similarities between the two loading rates during this phase. Unfortunately, direct µCT observations of the dynamic process at the temporal resolution required for comparison are not available to resolve this issue, for reasons stated in the introduction.

Local crack rotation with antithetic slip (Fig. 2c-iii) offers an additional mechanism for local stress rotation and slip on unfavourably-orientated faults without the need for high pore pressure[40]. It may also help to explain observations of interlaced orthogonal faults and accompanying geometric complexities (such as en-echelon faulting, aftershock migration and event triggering from one orientation to another) on a larger scale in the 2019 Ridgecrest earthquake sequence and other examples[41], by providing alternative pathways along which to accommodate strain during a cascading rupture process.

Our observations of crack rotation provide experimental evidence to support models of tectonic kinematics[42–45], which postulate that rotation in rifting margins and strike-slip settings can emerge as a result of local strength heterogeneity in the crust. In these cases, a strongly fixed zone (in our case, the rock matrix between en-echelon tensile microcracks) transmits the bulk drag, preventing pure shear and enabling extension and rotation, with rotation facilitated by the surrounding mechanically weak regions (in our case, the tensile cracks themselves) that accommodate the rotation-induced deformation. The rotation in our experiments is driven by shearing oblique to the en-echelon tensile cracks, whereby the strong grain segments transmit the shearing motion and prevent tensile crack propagation beyond a single grain, thereby limiting the width of the shear zone to the grain scale. Crack coalescence eventually occurs by grain fragmentation within the shear zone and by the breaking of asperities, forming strands of connected crack porosity aligned along the shear zone in patterns consistent with dynamic shear failure[46], coalescence of en-echelon rift segments[47] and kink band development in granodiorite[48]. The potential for slip along either and/or both surfaces of the shear zone, as well as planes within the shear zone, is also found on a larger scale in nature[49].

This discussion highlights the potential for re-examination of the microstructures and inferred mechanisms associated with larger-scale seismic and aseismic processes in light of the results presented here, in particular the conclusion that seismic events miss important grain-scale mechanisms governed by kinematics before and during shear failure. This improvement in process-based understanding holds out the prospect of reducing systematic errors in forecasting system-sized catastrophic failure in a variety of applications.

## Methods

For a description of how we implemented the AE feedback control of deformation, see Supplementary Note 1.

### Experimental material

A Permian aeolian sandstone from the Clashach quarry in Morayshire, Scotland was chosen as the experimental material. Clashach sandstone is a quartz-rich arenite composed of >92% quartz grains, <8% K-feldspar and subordinate lithics. It is well-sorted with fine to medium-sized grains 0.25–0.4 mm in diameter[50]. A highly cemented Clashach sample was used (17% porosity), which behaved in a mechanically brittle manner when loaded to failure at shallow crustal pressures, and emitted sufficient acoustic emissions for feedback control. Cylindrical cores of 10 mm diameter were obtained using a diamond core drill, and the ends of the cores were ground flat and parallel on a lathe to a length of 25 mm. This small sample size is required to obtain the micron-scale resolution achievable with synchrotron µCT imaging.

### Experimental equipment

The experiment used our lightweight (3.5 kg) X-ray transparent triaxial deformation apparatus, Stór Mjölnir (Supplementary Fig. 1), developed at the University of Edinburgh. Named after Thor's hammer in Norse mythology, it is an upscaled version of our miniature triaxial cell, Mjölnir[51], with the addition of a linear variable displacement transducer (LVDT) to measure axial displacement, and two piezoelectric P-wave transducers, positioned axially (Supplementary Fig. 1), to passively detect acoustic emissions (AE) and actively monitor ultrasonic velocities. These AE transducers were connected to a two-channel Applied Seismology Consulting Ltd (ASC) micro-seismic monitoring system by means of two ASC pre-amplifiers (full details given below in the description of acoustic emission recording and analysis). This allowed us to capture AE waveforms and conduct ultrasonic velocity surveys throughout the experiment. Stór Mjölnir is constructed of grade 5 titanium alloy, with an X-ray transparent pressure vessel made of 7068-T6 aluminium alloy. It accommodates cylindrical samples of 10 mm diameter and 25 mm length and can attain confining pressures up to 50 MPa and apply axial stresses up to 500 MPa. It was installed on the X-ray microtomography rotation stage in Experimental Hutch 1 (EH1) of beamline I12-JEEP at the Diamond Light Source.

### Experimental protocol

The experiment was conducted at ambient temperature. The Clashach core was jacketed in silicone tubing and sealed between the pistons within the pressure vessel (Supplementary Fig. 1). A high-viscosity, honey-like molasses (Sheargel by Charles Tennant & Co. Ltd.) was used as a couplant between the acoustic transducers and the pistons, and between the pistons and the sample ends to provide good transmission for both shear and compressional waves over a wide range of frequencies[52]. An effective pressure ($P_{eff}$) was applied and maintained at 20 MPa throughout the test (confining pressure $P_c = 25$ MPa and pore fluid pressure $P_p = 5$ MPa, where $P_{eff} = P_c – P_p$. A hydrostatic starting pressure condition (zero differential stress) was achieved by simultaneously increasing the axial pressure to match the confining pressure. Tomographic reference scans were acquired before pressurisation and again prior to loading (at zero differential stress) to obtain the initial state of the sample. Two scans were acquired at zero differential stress to characterise the error in the digital volume correlation by correlating two volumes in which the state of the sample was identical (Supplementary Fig. 7).

After the initial scans, the sample was loaded continuously at a constant deformation rate of $10^{-5} \, s^{-1}$ until the desired AE event rate ($1 \pm 1$ AE $s^{-1}$) was reached, at which point loading at this constant AE-rate took over. This AE event rate was established through extensive in-house testing, prior to the experimental campaign, to be the optimum

event rate for effective feedback control given the small sample size required for high-resolution μCT imaging (10 mm diameter × 25 mm length) and relatively few recorded AE events (~3500) compared with earlier AE feedback control experiments that used larger-scale samples. These earlier studies on granite[1,13,15] and sandstone[13,17,18] recorded a wide range of event numbers (4000–70,000) in samples of 50–76 mm diameter × 120–190 mm length. The samples used were dry, which is likely to increase the number and amplitude of events compared with a water-saturated sample (our case), but were conducted at higher confining pressure (40–50 MPa compared with 20 MPa in our case), which tends to suppress the number and amplitude of microcracks. The wide range of recorded events in these studies did not depend on rock type but may be related to the event rate used in each case, and, therefore, the duration of quasi-static shear zone development. It is difficult to compare our chosen optimum event rate with these studies since they do not state the AE event rate used. However, our protocol ensured that the AE feedback system took control of sample loading early enough to prevent dynamic failure but late enough to avoid a sudden increase in strain rate, and could be maintained throughout the failure. This enabled the failure time to be extended from ~1 to 50 min, equivalent to an average bulk strain rate of $10^{-7}$ s$^{-1}$, and sufficient to capture 18 high-quality μCT volumes after peak stress. The sample underwent triaxial deformation to failure evident from a rollover at peak differential stress followed by a gentle decrease in stress ending in almost constant stress on completion of loading (Fig. 1a and Supplementary Fig. 3).

Imaging the deformation in situ on beamline I12-JEEP[18] was achieved with a 53 keV monochromatic beam detected by a PCO.edge light sensor with I12 in-house optical module of 7.91 × 7.91 μm per pixel resolution and 20 × 12 mm field of view. Tomographic volumes of the whole sample, comprising two discrete overlapping scans of the top and bottom of the sample, with vertical translation in between, were acquired every 85 s throughout the experiment. Individual scans were acquired in ~40 s and consisted of 1800 projections with a 0.0035 s exposure time. At each end of the sample, 0.2 mm (next to the piston) was not captured due to limits on the X-ray field of view, which had a vertical dimension of 12 mm.

Ultrasonic velocity surveys were conducted every 5 min, and acoustic emissions monitored continuously throughout the experiment, along with actuator pressure and axial displacement, confining pressure, and pore fluid pressure and volume.

## Tomographic reconstruction

The tomographic dataset was fully reconstructed on the Diamond cluster using the framework SAVU[53] with automatic centre detection[54], ring removal[55] and distortion correction[56], a Paganin phase filter[57] with delta/beta ratio of 0.6, and a 2D GPU-accelerated filtered back-projection (FBP_CUDA) reconstruction algorithm using the integrated ASTRA toolbox[58–61]. Each reconstructed μCT volume was then cropped and down-sampled from 32-bit to 16-bit using grey-scale threshold limits of −0.00015 and 0.00055. This yielded 16-bit volumes of 1596 × 1506 × 1800 voxels. Each pair of reconstructed volumes comprising the top and bottom ends of the sample, acquired with an overlapping region of 520 vertical pixels (4 mm), was then merged together by digital volume correlation (DVC) using the open-source software SPAM[32] to match the overlapping ends and generate a time-series of μCT volumes of the full sample, each of 1596 × 1506 × 3080 voxels, with a voxel size of 7.91 × 7.91 × 7.91 μm$^3$ (orange dots in Fig. 1 and blue dots in Supplementary Fig. 3).

## Estimation of bulk stress and strain

Axial displacement was converted to axial strain, ε, which was then corrected for rig stiffness, $k_{rig}$ to obtain the axial sample strain, $\epsilon$, as follows: $\epsilon = \sigma(\varepsilon/\sigma - 1/k_{rig})$. Differential stress, σ, on the sample was obtained from $\sigma = \sigma_1 - \sigma_3$, where $\sigma_1$ is the axial stress, equal to the load,

F, on the sample (obtained from a calibration of the measured ram pressure with a load cell) divided by the surface area, A, of the sample end ($\sigma_1 = F/A$), and $\sigma_3 = P_{eff}$.

## Local incremental 3D strain fields from digital volume correlation

In addition to observing quasi-static fault development directly in the reconstructed μCT volumes, we quantified the full, incremental 3D strain tensor evolution using DVC between adjacent pairs of tomograms using the open-source software SPAM[32]. The procedure involved computing linear displacement vectors between nodes (with 40 pixel spacing, equivalent to 316.4 μm) identified in the reference image and the same nodes identified in the deformed image by tracking a representative volume (window size 40 pixels) around the nodes. The transformation gradient tensor (local derivatives of displacement), **F**, was then computed using a Q8 shape function linking 8 neighbouring nodes, with **F** computed in the centre of the element. **F** was decomposed into the symmetric stretch tensor, **U**, which was further decomposed into an isotropic and a deviatoric part. The volumetric strain (first invariant of the strain tensor showing dilation/compaction) was defined as the determinant of the transformation gradient tensor, $|\mathbf{F}| - 1$. Here, we followed the soil mechanics convention and defined dilation as positive volumetric strain and compaction as negative volumetric strain. The deviatoric strain (second invariant of the strain tensor showing shear deformation and referred to throughout as the shear strain) was defined as the Euclidean norm of the deviatoric part $\|\mathbf{U}_{dev}\|$.

## Dynamic wave velocity estimation

Changes in ultrasonic compressional-wave velocity were measured by means of the pulse transmission technique[62] between the pair of P-wave AE transducers, with source-receiver geometries at opposite ends of the sample (Supplementary Figs. 1 and 11). Velocity surveys were conducted by the ASC Pulser Interface Unit at constant time intervals (5 min) during the experiment. A pulse of acoustic energy was sent through the sample from each transducer in turn, and its arrival at the other sensor was recorded. To deduce the velocity change over time, the relative time delays between received signals were estimated by cross-correlation of successive arrival waveforms and corrected for the change in sample length over time. To understand the system delay (including travel time through pistons), a seismic test using the pistons without any sample in between them was recorded, and the difference in time between pulse onset at the origin and pulse onset retrieval at the end of the other piston was calculated as the time-correction.

## Acoustic emissions recording and analysis

Detected acoustic emission (AE) signals were received by the pair of Glaser-type P-wave piezoelectric transducers[63]. These transducers are broadband, with a flat response spectrum between 500 kHz and 2 MHz, and conical in shape, with a contact area 5 mm in diameter. The detected signals (in Volts and proportional to the true normal displacement of the received elastic wave) were amplified and sent first to the ASC Trigger Hit Counter, which triggered the recording of events based on an instantaneous amplitude threshold (trigger threshold), and then to the ASC Cecchi Acquisition Unit, which recorded the full event waveforms (at 50 MHz, 12 bit acquisition with 128 kilo-samples per channel), including arrival time, amplitude and first-motion information. The pre-amplification gain and trigger threshold were set at 70 dB and 280 mV, respectively, with the aim of detecting sufficient events above ambient noise to control the AE event rate effectively. The gain was determined from a benchmark pencil lead-break test in the laboratory prior to visiting the synchrotron, and the trigger threshold was determined by the ambient noise in the experimental hutch on the I12-JEEP beamline. Only AE events with amplitudes above the trigger threshold were recorded by the acquisition system, and AE

event rates were calculated from the number of events recorded in 10 s time intervals throughout the experiment. AE amplitudes as a function of time, AE event rate and frequency-magnitude distributions are shown in Supplementary Fig. 10.

We interrogated the AE events to obtain a sub-set of the most reliable events for location analysis, ensuring that: (i) individual events were represented in both transducers' recordings, (ii) the AE waveforms satisfied a minimum Signal to Noise Ratio (SNR), (iii) the relative time-delay for a specific AE recorded at the two transducers was robust against noise and yielded a depth uncertainty less than 1 pixel, (iv) the maximum relative time-delay for a specific AE recorded at the two transducers obeyed physical limits given sample dimensions/velocity. The relative time delays between the top and bottom transducers were estimated by cross-correlation of successive top and bottom waveforms, respectively. Given the limitation of only two AE transducers at the opposite ends of the rock sample, the experimental geometry led to inherent ambiguity in AE location. However, the specific relative time-delay of an AE event and knowledge of the dynamic velocity allowed us to define a circular hyperboloid, which marks a 3D surface of constant relative time-delay (Supplementary Fig. 11) and, therefore a kinematic constraint for AE location.

Using the AE events that passed the selection criteria (~5% of 3600 events) we defined each hyperboloid as the locus of potential elementary volumes (with thickness = 1 pixel) for the AE location. Time-binning of AE relative to the occurrence of the two tomograms used to obtain a particular strain increment from the DVC permitted comparison of AE with the local incremental strains. Assuming that each AE occurred at the largest incremental local strain within the kinematically defined hyperboloid, a unique AE location was established (location constrained by local strain). Given the large correlation between locations of large volumetric strain vs. deviatoric strain, AE locations were defined within the same pixel whether constrained by volumetric or deviatoric strain for 85% of the located AE. Where two or more AE had the same time-bin and the same kinematic constraint, we assumed that the largest AE would be correlated with the largest strain, effectively assigning AE locations sequentially from largest to lowest AE within a time-bin. Whilst the assumption of linking AE to the largest local strain should bias AE locations to be linked to larger local strains, the results showed that effectively via proof by contradiction, the largest AE do not occur at locations of large local strains, with the locations of large local strains being linked to small/moderate AE (Fig. 4). This observation, in turn, is indicative that deformation is primarily aseismic.

### Calibration of seismic moments and estimation of the overall seismic strain partition factor

We followed the approach of Dresen[3] et al., who estimated the seismic moment of the largest observed AE event based on the largest observed fault, in order to scale the distribution of seismic moments from all their events. We used the Kanamori and Anderson equation[64] to obtain the scalar seismic moments $M_O = CV\Delta\sigma$, with $C = 16/7$, $V = r^3$ the crack volume and $\Delta\sigma$ the stress drop. The estimated maximum scalar seismic moment was $1.5 \times 10^{-3}$ Nm, using the Madariaga source model[65] with a radius, $r$, equal to one-half the maximum observed crack size ($2r \sim 800$ μm) and a stress drop of 1 MPa. This stress drop lies in the middle of the range of stress drops seen in 23 different studies[66,67] of natural, mining-induced, and fracking-induced seismicity and laboratory AE events, with data spanning over 19 orders of magnitude in seismic moment (Supplementary Fig. 14). Our results are also consistent with published laboratory AE studies that use Glaser-type P-wave transducers, similar to ours (Supplementary Fig. 14, orange shaded area). The maximum amplitudes of the AE envelopes were scaled to the maximum scalar seismic moment to obtain the scalar seismic moment distribution. With this calibration, our events spanned a stress drop range of 0.1–10 MPa (straight, dashed lines in

Supplementary Fig. 14) similar to those seen at all scales in Supplementary Fig. 14.

The Kostrov strain[68] was estimated from $\triangle\boldsymbol{\varepsilon}_{ij} = \frac{1}{2\mu\triangle V}\sum_{i=1}^{N}\mathbf{M}_{ij}$, with moment tensor[69] $\mathbf{M}_{ij} = \sqrt{2M_0}\mathbf{U}_{ij}$, where $\mathbf{U}_{ij}$ is the unit displacement tensor, $\mu$ is the shear modulus and $\Delta V$ is the total representative volume. Finally, the strain partition factor[3,70,71] was estimated from $\chi = \frac{\gamma_{AE}}{\gamma_F}$, with $\gamma_{AE}$ representing the sum of scalar seismic moments multiplied by $\frac{\sqrt{2}}{2\mu\triangle V}$, and $\gamma_F$ representing the average volumetric and deviatoric strains accumulated throughout the deformation experiment.

### Estimation of bulk and local shear fracture energy

The breakdown zone (slip-weakening) model[27,28] (Fig. 6a) considers a fault as a shear crack with peak shear strength, $\tau_P$. Relative slip on the fault, $\Delta u$, initiates at $\tau_P$ and then strength degrades from $\tau_P$ to a constant residual frictional strength, $\tau_F$, as $\Delta u$ increases to a value $\Delta u^*$, the critical slip-weakening distance, where tensile fracturing (mode I; dilation) transitions to frictional sliding (mode II; shear). This leads to a breakdown zone of dimension $\omega_0$ at the shear crack tip. Shear fracture energy, $G_c$ for breakdown processes at the crack tip can be evaluated by integrating under the $\tau$-$\Delta u$ curve[27] from $\tau_P$ to $\tau_F$. $\Delta u$ is resolved from the bulk inelastic axial displacement, $\Delta l$, estimated from the differential stress vs axial displacement curve (see Fig. 1 in Wong et al.[28]) and depends on the angle, $\theta$, which the fault makes with the direction of loading. Shear stress depends on $\theta$ and the boundary stress conditions. Four main assumptions are involved: (i) all bulk axial strain beyond $\tau_P$ is accommodated by bulk shear slip on the fault, (ii) the fault is sample-sized throughout, (iii) slip is uniformly distributed across the whole fault, and (iv) $\omega_0$ is small compared with the fault size.

To obtain local geometric and strain information for the shear zone itself—defined by the local strain fields that were output from the DVC (Fig. 3 and Supplementary Fig. 4)—we separated the shear zone core from other correlated strain clusters by thresholding the shear strain fields above 0.004 strain. This threshold was chosen at a level where the shear zone was well-defined as a coherent object, even though it was still somewhat patchy in the dilation field. We then used a connected component object analysis to obtain the best-fitting ellipsoid for the shear zone at each strain increment, and hence obtained the angle $\theta$ of the shear zone with respect to the direction of loading (Supplementary Table 1) from the ellipsoid eigenvectors. The evolving shear zone object is shown in Supplementary Fig. 9.

Following the slip-weakening model[27,28], we estimated $G_c = \int_0^{\triangle u^*}[\tau(\triangle u) - \tau F]\, d(\triangle u)$ for:

(i) Uniform slip on a sample-sized fault[28] by first resolving relative slip, $\Delta u$, from the bulk inelastic axial displacement, $\Delta l$, and $\theta$ as follows: $\Delta u = \Delta l/\cos\theta$, and then calculating shear stress, $\tau = [(\sigma_1 - \sigma_3)/2] * \sin(2\theta)$, from $\theta$ and the boundary stress conditions. A cubic fit was used to obtain an average bulk shear stress vs relative slip curve, and we assumed that all axial shortening was caused by slip along the fault from $\tau_P$. Axial shortening was corrected for elastic strain using the intact Young's modulus for the sample (Figs. 1c and 5a).

(ii) Uniform slip at each μCT scan time, using the actual bulk shear strain and relative slip measurements at those specific times (rather than an average), in order to compare the uniform slip model directly with the observed local slip on the shear zone itself (estimated from the 3D strain fields). We assumed that all axial shortening was caused by slip along the fault from the onset of microcrack localisation along the critically oriented shear zone (tomogram 22).

(iii) Total observed local slip on the shear zone itself from local dilation and shear strain. We assumed that slip started from the onset of microcrack localisation along the critically oriented shear zone (tomogram 22) and used the bulk shear strain calculated in (ii). For the shear contribution to slip, we extracted the mean local

incremental shear strain in the shear zone object, $\overline{\triangle\epsilon_{shear}}$, from the connected component object analysis described above. We corrected the cumulative local shear strain, $\sum\overline{\triangle\epsilon_{shear}}$, for elastic strain using the intact shear modulus of the sample, obtained from the shear stress vs $\sum\overline{\triangle\epsilon_{shear}}$ curve (Supplementary Fig. 8). Finally, we obtained local shear slip, $\triangle u_{shear}$, from engineering strains approximation[72] and the DVC node spacing, $n$: $\triangle u_{shear} = 2n\sum\overline{\triangle\epsilon_{shear}}$. For the radial dilation contribution to slip, we extracted the mean local incremental radial dilation in the shear zone object (the extent of which was defined from the shear strain fields), $\overline{\triangle\epsilon_{dilation}}$, and corrected the cumulative local dilation, $\sum\overline{\triangle\epsilon_{dilation}}$, for elastic radial strain using the elastic axial strain calculated above and an estimate of Poisson's ratio for Clashach sandstone[73]. We then resolved the inelastic dilation onto the shear zone orientation, obtaining local dilational slip from an engineering strains approximation[72]: $\triangle u_{dilation} = n\sum\overline{\triangle\epsilon_{dilation}}/\sin\theta$.

## Data availability

The reconstructed 3D X-ray µCT volumes and 3D strain fields, along with the raw acoustic waveforms and raw mechanical data, generated in this study are available at the NERC EDS National Geoscience Data Centre repository under accession code 173296 https://doi.org/10.5285/56c7802c-93db-4f0f-8b89-e18e10215633, with direct access to the dataset available at the NERC STFC Centre for Environmental Data Analysis third party holdings link [https://data.ceda.ac.uk/ngdc/NE_R001693_1]. These data are available under Open Government Licence (OGL). When using the dataset held in the repository, please cite Cartwright-Taylor et al.[74]. The raw X-ray µCT radiograph files are very large and are stored at the Diamond Light Source. They are available on request from the authors. All other processed data supporting our conclusions can be found in the main manuscript and in the Supplementary Information.

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

## Acknowledgements

This work is supported by the UK's Natural Environment Research Council (NERC) through the CATFAIL project NE/R001693/1 *Catastrophic failure: what controls precursory localisation in rocks?* (Principal Investigator I.G.M.). We acknowledge Diamond Light

Source for time on beamline I12-JEEP under proposal MG22517 (Principal Investigator I.B.B.). We would also like to thank the University of Edinburgh Geosciences Workshop for their support in developing the experimental apparatus, and Jonathan Singh for useful discussions about data file formats and estimation of acoustic waveform arrival times.

## Author contributions

Contributions are presented according to the CRediT model. A.B., I.B., A.C.-T., A.C., F.F. and I.M. contributed to the conceptualisation of the research and the acquisition of funding. A.C.-T. and M.-D.M. were responsible for the data curation and visualisation. E.A., A.B., A.C.-T., A.C. and M.-D.M. undertook the formal analysis. I.B., A.C.-T., F.F., D.L., O.M., S.M. and R.R. conducted the experimental investigation. E.A., A.B., I.B., A.C., A.C.-T., F.F., M.L., I.M. and M.-D.M. designed the methodology. A.C.-T. and I.M. were responsible for the project administration. E.A., I.B., M.L. and O.M. provided resources. E.A. and M.L. developed software. A.B., I.B., A.C., F.F. and I.M. provided supervision. A.C.-T., A.C., I.M. and M.-D.M. undertook validation. A.C.-T., M.L., I.M. and M.-D.M. wrote the original draft of the manuscript. All authors reviewed and edited the manuscript.

## Competing interests

The authors declare no competing interests.
