## [Peer Review File · Nature Communications]

REVIEWER COMMENTS

Reviewer #1 (Remarks to the Author):

This paper presents the results of a single and unique triaxial rock deformation experiment that was conducted in situ at a synchrotron beamline. The experiment was specifically designed so as to be able to image the microstructure evolution during dynamic failure using X-ray tomography provided by synchrotron radiation coupled with acoustic monitoring of the sample. To the extent of my knowledge on the topic, this is an unprecedented study. I am delighted by its high quality, the amount of work that was performed, and the richness of the results. Although one could argue that this is a single experiment and probably warrants repetition and further confirmation, I would argue that this kind of experiment is challenging enough to be admirative of the outcome. Bravo!

I have read the manuscript carefully, scrutinised the methods and gone through the supplementary material, and found no major flaw. The work is solid and the results sound. I, therefore, won't provide any major discussion point as I am very convinced by all the results and interpretations put forward. I do, however, have a few minor points to make which are listed below and one suggestion for a figure. Once these have been addressed, I will be more than happy if this piece of work was to be accepted.

On a personal level, it is very satisfying to see all those grain-scale mechanisms illuminated by X-ray tomography as they reveal that a lot more aseismicity is associated with strain localisation than I (and probably many others) would have anticipated. It is also as equally satisfying to see how these tomographic results helps estimating the work associated with those aseismic mechanisms, which accounts for up to 40% of the total shear fracture energy (as classically estimated from bulk stress-strain data).

Figure suggestion

Given that the dynamic wave velocity inside the sample is measured every five minutes during the experiment, why not having its temporal evolution plotted somewhere? I would be curious to see if it follows what is expected, i.e. a decrease through time due to microcrack nucleation, coalescence and strain localisation onto the macroscopic shear fault plane. Also, it would inform the AE location results shown in Supplementary Figure 12.

Minor points

i) Line 315 – please double check the equation's consistency as the first term on the right-hand side (i.e., ϵ/σ) has dimensions of an inverse pressure when it should be dimensionless.

ii) Line 387 – you mean Supplementary Figure 14 and not 13, I believe.

iii) Line 626 – should read '(c)' and not '(b)'.

iv) Supplementary Figure 10c – is this the frequency-magnitude distribution for all the events recorded throughout (i.e., when bulk sample failure ensues at about 6000 s)? If so, please specify it.

Reviewer #2 (Remarks to the Author):

Dear authors,

Please find below my feedback. I believe that this is a very good piece of work that deserves publishing in Nature. I would, however, like to ask you further elaborate on my comments below.

General overview

Cartwright-Taylor and co-workers present results from a triaxial compression experiment with Acoustic Emission (AE) feedback control performed on a saturated sandstone specimen. Previous authors (Lockner and co-workers (1991); Stanchits et al., 2006, 2011, Charalampidou et al., 2014 (EGECE)) performed triaxial compression experiments with AE feedback control of the loading rate, when squeezing much larger, however, specimens (40-50 mm in diameter). During these studies the lab-induced deformation micro-processes were studied by means of AE locations and source mechanisms (first motion polarities and/or Moment Tensor inversion).

The novelty of the work by Cartwright-Taylor and co-workers is that AE feedback control of the loading rate was applied to much smaller sample (10 mm in diameter) during the triaxial compression experiment. Moreover, the experimental set-up used in their study was also coupled with syn-deformation x-ray CT (performed at the I12-JEEP beamline at the Diamond Light Source). An upgraded version the x-ray transparent rock deformation cell, Stór Mjöllnir, with acoustic monitoring (which, for the moment, is quite unique) was used for the purposes of their work. The authors comment on the micro-processes based on the observed textural changes linked to the lab-induced deformation (x-ray CT data), the calculated shear and volumetric strain fields (Digital Volume Correlation (DVC)) and the AE locations (using only 2 vertically oriented P-wave transducers). AE locations usually require at least 6 piezoelectric sensors, which provide good sample coverage and they are mainly attached to/glued on the periphery of the tested specimen. The concept of using only 2 sensors to locate AE events is a considerable contribution when combining, however, acoustic with syn-deformation x-ray CT measurements. In terms of the micro-mechanisms, syn-deformation x-ray CT and DVC have demonstrated shearing and dilation along the main micro-fault (after the peak stress) and several textural damage-related mechanisms were identified by the X-ray images during the experiment (e.g., tensile cracks, grain crushing and pore collapse (cataclasis), rotation of grain fragments, grain fragmentation etc).

One of the main observations based on the calculated strain fields and the first-motion

amplitude of the AE locations (Fig. 4 and within the text) is that higher amplitude AE events occur in places that are not characterised by the highest shear and volumetric strains. Thus, smaller amplitude events occurred in regions of high local strain suggesting, according to the authors, that deformation in the shear zone is primarily aseismic. I would like to raise some concerns with the current interpretation, which I discuss below.

First, I will provide some answers to the questions requested for the review. These will be followed by some further comments to the authors.

Review questions

- What are the noteworthy results? Please, see above.
- Will the work be of significance to the field and related fields? How does it compare to the established literature? If the work is not original, please provide relevant references.
Yes, I believe that the work is of great significance in the field. I believe the authors need to further discuss their results with existing literature (i.e., other studies on larger samples, which also used AE feedback applied to loading rate, e.g., Lockner and co-workers (1991); Stanchits et al., 2006, 2011, Charalampidou et al., 2014).
- Does the work support the conclusions and claims, or is additional evidence needed?
I would like to ask the authors to further elaborate on the link between AE amplitudes and magnitude of shear and volumetric strains – please see further suggestions below. The outcome itself is good, but I would be more confident if extra feedback is provided to the further suggestions/ comments.
- Are there any flaws in the data analysis, interpretation, and conclusions? Do these prohibit publication or require revision?
I believe that this work should be published in Nature. This is a unique set of data; it is a difficult experiment, and the wealth of the acquired data is considerable. As such, I strongly believe that this experiment can provide experimental evidence that will increase our understanding of the occurring micro-processes during failure in saturated sandstones. However, I would like to ask for some further elaboration on the interpretation of the results (see further comments).
- Is the methodology sound? Does the work meet the expected standards in your field?
Yes.
- Is there enough detail provided in the methods for the work to be reproduced?
Yes, some extra information required is summarised below.

Further comments

Line 39: Is this uniaxial or triaxial compression? I think it is triaxial.

Lines 64-65: 'Mapping AE source locations during such an experiment provided the first in-situ view of crack localisation and shear zone growth' this is not the first in-situ view. Work has been previously done by Lockner and co-workers 1991, 1992, Stanchits et al, 2011 and Charalampidou et al., 2014. What you can very nicely mention here is that this is the first time a lab-induced shear zone was mapped in small-scale samples, and in that sense, it would be very interesting to compare any differences or similarities in the observed micro-processes

within the different size samples, i.e., compare your AE locations (onset and propagation) with those observed in the above studies in larger sandstone samples.

Figure 1: Why have you selected this AE rate (1AE/sec)? Elaborate a bit further on the selection of this threshold. Figure caption: please change second (1b) to (1c). How have you defined the yield point? How the time intervals (i-v) in (1a) correlate with strain intervals (a-g) in (1c)? You can potentially add the (i-v) vertical lines to (1c). Have you thresholded any of the AE recorded amplitudes or do you present all AE recorded amplitudes in (1a)?

Figure 2: Add the resolution of the XRT images (scale) somewhere in the picture. Add the movement of the piston (upwards or downwards, using an arrow indicating the applied global force). In (2aii) I can see the movement of the hanging wall, but, you can, possibly, expand laterally the white dashed line so your readers can see the vertical boundaries of your vertical cross section. In (2aiii) the white dashed line on the shear band covers the grain texture you want to visualise. I have 2 suggestions here: a) show a zone instead of the core (i.e., two offset lines, similar to what you show at (2aii), but due to the small size of the image one cannot see any grain scale activity clearly); b) show the vertical displacement field of this cross section (see Charalampidou et al., 2014 (IJRMMS)) – in that case, I would use a time interval that is closer to your end-product (XRT image you visualise here); this can be either the displacement field calculated between 1st and last XRTs or any smaller time interval you have calculated and can be of interest. I'm pretty sure that the displacement field can also capture the grain crushing shown on the bottom left side of the current vertical cross section. (2aii). Moreover, please, elaborate further on the time-lapse histogram. How have you defined the partial volume region and the rock matrix region? What covers the in-between them space on the histogram? In (2aiii), what do the different identified mechanisms tells us about the processes? Can you also comment on the role of local porosity that facilitates or halts some mechanisms? You have very good resolution XRT images for that purpose. I'm basically suggesting a couple of sentences that relate the observed mechanisms with the initial pore space (that evolves as well) with deformation.

Figure 3: How have you defined a) yield; b) final localisation (?) – maybe you need to find another term here; something like pre-stress drop or something similar; same for the other terms – is it shear zone propagation or better development? I think that the development starts from interval c; then, how are you sure that 'coherent sliding' is not taking place at the top part of the shear band also in what you call propagation interval (based on visual inspections of your incremental strain fields)? I believe that if you well define this terminology, then it is fine to go with it. You might also like to further talk about the observed microprocesses that potentially coexist at different patches of your evolving deformation band during subsequent time intervals.

Lines 207-217: Which Figure supports the indicates numbers for the fracture energy? It is a bit unclear to me whether in this paragraph you talk about the energy budget in general (so fracture, radiated and heat energies) or only the fracture energy. You might also need to further elaborate on the seismic and aseismic classification. If this paragraph is on energy budget in general, what kind/type of uncertainties you anticipate with regards to the radiated energy given that you have worked with only 2 sensors and given that you considered only the 5% of the recorded AE?

Lines 221- 222: Local crack rotation with antithetic slip (Fig 2c-iii) offers an additional

mechanism for local stress rotation. How frequent this is within your sample? Rotation can be accommodated where the local dilation allows it. Is it something you observe frequently in the x-ray images?

Lines 242-245: This discussion highlights the potential for re-examination of the microstructures and inferred mechanisms associated with larger-scale seismic and aseismic processes considering the results presented here, in particular, the conclusion that seismic events miss important grain-scale mechanisms governed by kinematics before and during shear failure. Do you believe that the occurring micro-processes would have been the same if you did not apply the AE feedback? How the AE feedback control relates to larger-scale seismic and aseismic processes?

Line 269: Can you mention the gain of the pre-amplifiers used, also in the main text? Can you also provide some further details on the frequency of the PZE and potentially the width of the crystal (or some references)? Please add to the main text that these are P-wave sensors.

Line 329: Can you provide further details about the 2-channel monitoring system (model etc)?

Line 364: SPAM – does the software consider the grain rotation when calculating the strain fields? Or else, do you have any calculated rotation field?

Line 385-387: a stress drop of 1 MPa; The latter stress drop is an intermediate stress drop encompassing 23 different studies of natural, mining induced, and fracking induced seismicity and laboratory AE events. How this stress drop number relates to your experiment? Why such a stress drop? Please provide some comments to link this with your experimental data.

Lines 404-405: Given that the DVC has revealed strains not in a single plane (local variability, off plane volumetric and shear strains) can you elaborate further one of the assumptions, i.e., all axial strain is accommodated by shear slip in the fault plane (beyond peak)?

Figure 4 (and equivalent text)

I have gathered here several parts of the text, where you discuss Figure 4.

Lines 116-118: These observations highlight the significant contribution of aseismic mechanisms (i.e., rotation) to the overall failure process, relative to seismic mechanisms (i.e., cracking, stick-slip sliding; 6).

Lines 139- 142: Many small events occurred in regions of high local strain (Fig. 4 dashed black ellipses), consistent with deformation being primarily aseismic in the shear zone, while many large events occurred in regions of low local strain in the bulk (Fig. 4 and Supplementary Figure 12).

Lines 166-167: For the AE location (definition of the unique position of the AE) you assume that each AE occurred at the largest local (shear and/or volumetric) strain within its circular hyperboloid.

Line 198: moderate/small events occurring in regions of large directly measured strain

Lines 202-203: therefore, the AE source amplitude is not necessarily representative of the local strain

Lines 354-357: Given the large correlation between locations of large volumetric strain vs. deviatoric strain, AE locations were defined within the same pixel whether constrained by volumetric or deviatoric strain for 85% of the located AE.

Lines 360-363: Whilst the assumption of linking AE to the largest local strain should bias AE

locations to be linked to larger local strains, the results showed that, effectively via proof by contradiction, the largest AEs do not occur at locations of large local strain, with the locations of large local strains being linked to small/moderate AE (Fig. 4). This observation, in turn, is evidence that deformation is primarily aseismic.

My comments: Fig 4 shows (for the 5% of the recorded AE events) that larger (higher amplitude) AE events occurred in regions of lower local shear and volumetric strains whereas smaller AE events occurred in regions of higher local shear or volumetric strains. The authors suggest that the latter is consistent with deformation being primarily aseismic in the shear zone.

In my opinion another explanation for the above observation can be the type of the AE events (or else their source mechanisms). Personal experience indicates that shear and tensile type events have usually lower amplitudes compare to compressive type events. Moreover, in saturated samples, signals are usually weaker, so you might have lost a big number of events also because of that. Going back to my source mechanisms argument, I'm expecting shear and tensile type events in regions that are characterised by high shear and high positive volumetric strain (i.e., dilatant patches). So, this means that lower amplitude AE events (shear and tensile) co-exist with high shear and high volumetric (dilatant strains) – which is what you have in Fig 4. Compressive events (pore collapse and/or grain crushing) have larger amplitudes and I expect them to coincide with lower shear strains and higher volumetric (compactant) strains. Usually, the volumetric strains are not well resolved (at least as well as the shear strains). Moreover, you might have a compressive event (grain crushing or pore collapse) that generates a larger amplitude AE, but it is so local that is not resolved within your DVC window (or representative volume = 40 pixels x 7-8 um= 320 um). Q1: Haven't you observed any compactant (negative according to your convention) strains within those regions included in Fig. 4? And if yes, how do they correlate with the higher amplitude events? Q2: Are all volumetric strains in Fig 4 dilatant or are you plotting the absolute values of the volumetric strains? Have you divided with the absolute volumetric max? I would plot separately compactant and dilatant volumetric strains (and I would also divide each group with the max compactant or max dilatant). Q3: Have you considered any uncertainties related to what is visualised by the DVC because of limitations of the method? Here, I refer to case of compressive events (so larger amplitudes) and low compactant strains (shear strains are expected to be low here anyway).

Finally, in terms of processes, aseismic deformation involves rotations and large enough dilation to have floating fragments that rotate without getting in touch with other grains and cements. Any contact of the rotating fragment with other grains will generate further deformation and thus, will radiate acoustic energy, so it will not be aseismic anymore. How many of the above cases have you identified within your x-ray CT images? Is this a common (see also % of aseismic share you suggest in the energy budget session) observation? So, what I am suggesting here is to explore further the above argument and if this does not hold for your datasets, to continue with your existing arguments of aseismic deformation.

Response to reviewers

We thank both reviewers for their positive comments and thoughtful suggestions below. Each point (in **black** text) has been addressed individually in **orange** text, with changes to the manuscript text shown in **purple**. Unchanged manuscript associated with these changes is shown in **blue** (as here). Changes in the manuscript itself are shown underlined in **red** text.

Reviewer 1

This paper presents the results of a single and unique triaxial rock deformation experiment that was conducted in situ at a synchrotron beamline. The experiment was specifically designed so as to be able to image the microstructure evolution during dynamic failure using X-ray tomography provided by synchrotron radiation coupled with acoustic monitoring of the sample. To the extent of my knowledge on the topic, this is an unprecedented study. I am delighted by its high quality, the amount of work that was performed, and the richness of the results. Although one could argue that this is a single experiment and probably warrants repetition and further confirmation, I would argue that this kind of experiment is challenging enough to be admirative of the outcome. Bravo!

I have read the manuscript carefully, scrutinised the methods and gone through the supplementary material, and found no major flaw. The work is solid and the results sound. I, therefore, won't provide any major discussion point as I am very convinced by all the results and interpretations put forward. I do, however, have a few minor points to make which are listed below and one suggestion for a figure. Once these have been addressed, I will be more than happy if this piece of work was to be accepted.

On a personal level, it is very satisfying to see all those grain-scale mechanisms illuminated by X-ray tomography as they reveal that a lot more aseismicity is associated with strain localisation than I (and probably many others) would have anticipated. It is also as equally satisfying to see how these tomographic results helps estimating the work associated with those aseismic mechanisms, which accounts for up to 40% of the total shear fracture energy (as classically estimated from bulk stress-strain data).

Figure suggestion

Given that the dynamic wave velocity inside the sample is measured every five minutes during the experiment, why not having its temporal evolution plotted somewhere? I would be curious to see if it follows what is expected, i.e. a decrease through time due to microcrack nucleation, coalescence and strain localisation onto the macroscopic shear fault plane. Also, it would inform the AE location results shown in Supplementary Figure 12.

We have included a new figure in the main text showing the evolution of velocity as measured from the active seismic tests, alongside the differential stress evolution. We have also included cross-plots of deviatoric vs. volumetric strain, to provide an understanding of the deformation processes leading to velocity changes. This is now presented as a new Figure 4, with the following caption: **Fig. 4. Velocity, stress and AE event rate evolution as a function of time, with selected incremental strain field cross-plots.** (a) Velocity evolution, defined from active seismic surveys, using the top sensor as a receiver with the bottom sensor as the seismic source (blue line) vs. the bottom sensor as a receiver with the top sensor as the seismic source (red line). (b) Differential stress throughout loading (blue line) along with AE event rate (green line). Orange lines denote the times of the μ CT volumes. (c-g) Normalised cross-plots of the local incremental deviatoric (shear) strain, $\Delta\epsilon_{deviatoric}$, vs. incremental volumetric strain, $\Delta\epsilon_{volumetric}$ showing the initial transition from (c) compaction-dominated strain to (d) dilation-dominated strain, (e-f) the correlation between dilation and shear strain through localisation and shear zone development and (f) the relaxation of dilation once the shear band is fully developed, and finally (g) shear-enhanced compaction during coherent sliding. Dilation is defined as positive volumetric strain and number labels refer to the

position of the strain increment in the time-series (as in Fig. 1a). The full time-series of cross-plots is shown in Supplementary Fig. 16.

We have also included a description of the velocity evolution in the text as follows (lines 197-211):

Ultrasonic velocity surveys, with source-receiver geometries at opposite ends of the sample (Supplementary Fig. 1), were performed every five minutes throughout the experiment to characterise the compressional-wave velocity (V_p) along the loading direction, and to locate the acoustic emissions (AE). Figure 4 shows that V_p initially increased in response to compaction (aforementioned stage i) and then decreased during strain hardening (stage ii), in line with the observed transition from compaction-dominated to dilation-dominated local strain at the yield point, and concurrently with exploratory strain localisation (Fig. 3). Beyond peak stress, strain softening (stage iii) and shear zone propagation (stage iv) were marked by a continued decrease in V_p due to dilatant microcracking and newly generated pore space along the shear zone associated with localised dilation and shear strain (Figs. 2, 3 and 4). V_p continued to decrease throughout coherent slip (stage v; tomograms 31-end), although at a slightly reduced rate, indicating continued but reduced dilation as the local deformation mechanism became predominantly shear with an increased contribution from compaction (Fig. 4), consistent with previous observations of AE source types^{6,16-18}. However, V_p never recovered to its original value, indicating greater early compaction than subsequent dilation along the direct arrival path of the axial P-wave.

We also discuss the velocity evolution in context with previous studies as follows (lines 273-286): The axial V_p evolution is in agreement with published laboratory measurements^{13,15-18}. Our combined dataset confirms previous inferences from independent AE studies regarding the micro-mechanisms involved. The initial increase in velocity occurs in response to pore size reduction and closure of intra- and inter-granular cracks and high aspect-ratio pore spaces, which leads to a reduction in excess compliance²⁹. In response to a competing mechanism, dilatant micro-cracking, which tends to reduce the velocity, there is a non-linear reduction in the rate of V_p increase until dilatant micro-cracking dominates at the yield point and the velocity shows an overall reduction thereafter. In our case, V_p never recovered to its original value, consistent with earlier observations¹⁶⁻¹⁸. This reflects the increasingly heterogeneous damage evolution, whereby axial V_p is less sensitive to the mainly radial dilation that occurs during tensile microcracking, consistent with observations of increasing ultrasonic velocity anisotropy in larger samples¹⁵⁻¹⁸. The difference from earlier studies is that we have independently verified the underlying mechanisms of compaction and dilatancy from the strain fields, hence validating hypotheses derived from these earlier studies.

Minor points

- i) Line 315 – please double check the equation's consistency as the first term on the right-hand side (i.e., ϵ/σ) has dimensions of an inverse pressure when it should be dimensionless. Thanks, this is a typo. Equation changed to: $\epsilon = \sigma[\epsilon/\sigma - 1/k_{rig}]$ (line 524)
- ii) Line 387 – you mean Supplementary Figure 14 and not 13, I believe. Yes – changed (line 608).
- iii) Line 626 – should read '(c)' and not '(b)'. Changed (line 887)
- iv) Supplementary Figure 10c – is this the frequency-magnitude distribution for all the events recorded throughout (i.e., when bulk sample failure ensues at about 6000 s)? If so, please specify it. We have added the following text to the caption: (c) Frequency-magnitude plot showing incremental (blue) and cumulative (orange) distributions for all detected events throughout the experiment (i.e., all those detected by 6000 s at the point of bulk sample failure).

Reviewer 2

Dear authors, please find below my feedback. I believe that this is a very good piece of work that deserves publishing in Nature. I would, however, like to ask you further elaborate on my comments below.

General overview

Cartwright-Taylor and co-workers present results from a triaxial compression experiment with Acoustic Emission (AE) feedback control performed on a saturated sandstone specimen. Previous authors (Lockner and co-workers (1991); Stanchits et al., 2006, 2011, Charalampidou et al., 2015 (EJECE)) performed triaxial compression experiments with AE feedback control of the loading rate, when squeezing much larger, however, specimens (40-50 mm in diameter). During these studies the lab-induced deformation micro-processes were studied by means of AE locations and source mechanisms (first motion polarities and/or Moment Tensor inversion).

The novelty of the work by Cartwright-Taylor and co-workers is that AE feedback control of the loading rate was applied to much smaller sample (10 mm in diameter) during the triaxial compression experiment. Moreover, the experimental set-up used in their study was also coupled with syn-deformation x-ray CT (performed at the I12-JEEP beamline at the Diamond Light Source). An upgraded version the x-ray transparent rock deformation cell, Stór Mjöllnir, with acoustic monitoring (which, for the moment, is quite unique) was used for the purposes of their work. The authors comment on the micro-processes based on the observed textural changes linked to the lab-induced deformation (x-ray CT data), the calculated shear and volumetric strain fields (Digital Volume Correlation (DVC)) and the AE locations (using only 2 vertically oriented P-wave transducers). AE locations usually require at least 6 piezoelectric sensors, which provide good sample coverage and they are mainly attached to/glued on the periphery of the tested specimen. The concept of using only 2 sensors to locate AE events is a considerable contribution when combining, however, acoustic with syn-deformation x-ray CT measurements. In terms of the micro-mechanisms, syn-deformation x-ray CT and DVC have demonstrated shearing and dilation along the main micro-fault (after the peak stress) and several textural damage-related mechanisms were identified by the X-ray images during the experiment (e.g., tensile cracks, grain crushing and pore collapse (cataclasis), rotation of grain fragments, grain fragmentation etc).

One of the main observations based on the calculated strain fields and the first-motion amplitude of the AE locations (Fig. 4 and within the text) is that higher amplitude AE events occur in places that are not characterised by the highest shear and volumetric strains. Thus, smaller amplitude events occurred in regions of high local strain suggesting, according to the authors, that deformation in the shear zone is primarily aseismic. I would like to raise some concerns with the current interpretation, which I discuss below. First, I will provide some answers to the questions requested for the review. These will be followed by some further comments to the authors.

Review questions

- What are the noteworthy results?

Please, see above.

- Will the work be of significance to the field and related fields? How does it compare to the established literature? If the work is not original, please provide relevant references.

Yes, I believe that the work is of great significance in the field. I believe the authors need to further discuss their results with existing literature (i.e., other studies on larger samples, which also used AE feedback applied to loading rate, e.g., Lockner and co-workers, 1991; Stanchits et al., 2006, 2011, Charalampidou et al., 2015).

- Does the work support the conclusions and claims, or is additional evidence needed?

I would like to ask the authors to further elaborate on the link between AE amplitudes and magnitude of shear and volumetric strains – please see further suggestions below. The outcome itself is good, but I would be more confident if extra feedback is provided to the further suggestions/ comments.

- Are there any flaws in the data analysis, interpretation, and conclusions? Do these prohibit publication or require revision?

I believe that this work should be published in Nature. This is a unique set of data; it is a difficult experiment, and the wealth of the acquired data is considerable. As such, I strongly believe that this experiment can provide experimental evidence that will increase our understanding of the occurring micro-processes during failure in saturated sandstones. However, I would like to ask for some further elaboration on the interpretation of the results (see further comments).

- Is the methodology sound? Does the work meet the expected standards in your field? Yes.

- Is there enough detail provided in the methods for the work to be reproduced? Yes, some extra information required is summarised below.

Further comments

Line 39:

Is this uniaxial or triaxial compression? I think it is triaxial. Yes, added (line 40): triaxial

Lines 64-65:

‘Mapping AE source locations during such an experiment provided the first in-situ view of crack localisation and shear zone growth’ this is not the first in-situ view. Work has been previously done by Lockner and co-workers 1991, 1992, Stanchits et al, 2006, 2011 and Charalampidou et al., 2015. What you can very nicely mention here is that this is the first time a lab-induced shear zone was mapped in small-scale samples, and in that sense, it would be very interesting to compare any differences or similarities in the observed micro-processes within the different size samples, i.e., compare your AE locations (onset and propagation) with those observed in the above studies in larger sandstone samples.

The sentence starting ‘Mapping AE source locations during such an experiment provided the first in-situ view of crack localisation and shear zone growth’ (line 66) is not a claim of the paper, but is part of the literature review. In this sentence we are referring to the Lockner 1991 and 1992 studies (refs. 1 and 13, referred to at the end of the sentence), which did provide the first in-situ view of microcrack localisation and fault growth by mapping AE source locations during an AE-rate controlled experiment. We have amended the text for clarity, and to include the additional references (thank you for pointing those out) as follows (lines 66-72): Mapping AE source locations during such an AE-rate controlled experiment^{1,13} provided the first *in-situ* view of microcrack localisation along a shear zone and subsequent shear zone growth by continued microcracking, as well as an estimate of the associated shear fracture energy, a key parameter in the mechanics of earthquakes and faulting. Since then, several *in-situ* acoustic monitoring studies of loading rate effects¹⁵ and AE source mechanisms¹⁶⁻¹⁸ in rocks undergoing deformation have controlled the applied load to maintain a constant AE event rate and slow down the failure process.

We have added the following to the discussion to clarify our claim (lines 254-255): Our experiment provides the first integrated view of crack localisation and shear zone development, combining both *in situ* x-ray μ CT and *in situ* acoustic data.

While our seismic partition coefficient is estimated on the full set of recorded AEs (3600 events), comparable to some laboratory rock deformation studies (Lockner et al., 1992; Charalampidou et al., 2015) on much larger samples (40-75 mm diameter compared to our 10 mm diameter sample), our location results are for a

very limited dataset (~200 events) which were time-resolved into the 36 time-steps of the incremental 3D strain fields. These AE represented the largest events with most robust kinematic signatures in terms of small uncertainty in position. As such, our location analysis is not directly comparable to studies involving AE location (e.g., Lockner et al., 1991; 1992; Stanchits et al., 2006, 2011; Charalampidou et al. 2015), which analyse the spatio-temporal evolution of thousands of events. For the limited number of located events, there is an overall trend of events being more distributed throughout the sample during the early stages of deformation (strain increments 1-19) and for smaller events to localise (strain increments 19 onwards) along the three candidate shear zones and, eventually, the critically-oriented shear zone. These findings, although based on the limited set of located AEs, are consistent with the aforementioned studies which comment on a dispersed cloud of events eventually localising onto distinct deformation planes.

We have added the following discussion linking inferences from our AE data to deformation processes, in the context of previous studies:

Lines 255-259: It validates many inferences from classic AE experiments, such as the nucleation and growth of a shear zone containing the eventual fault plane due to the spontaneous localisation of en-echelon tensile (dilatant) microcracks^{1,13}. It also demonstrates that shear and compactant micro-mechanisms become increasingly important during shear zone development^{6,16-18}.

Lines 273-303:

The axial V_p evolution is in agreement with published laboratory measurements^{13,15-18}. Our combined dataset confirms previous inferences from independent AE studies regarding the micro-mechanisms involved. The initial increase in velocity occurs in response to pore size reduction and closure of intra- and inter-granular cracks and high aspect-ratio pore spaces, which leads to a reduction in excess compliance²⁹. In response to a competing mechanism, dilatant micro-cracking, which tends to reduce the velocity, there is a non-linear reduction in the rate of V_p increase until dilatant micro-cracking dominates at the yield point and the velocity shows an overall reduction thereafter. In our case, V_p never recovered to its original value, consistent with earlier observations¹⁶⁻¹⁸. This reflects the increasingly heterogeneous damage evolution, whereby axial V_p is less sensitive to the mainly radial dilation that occurs during tensile microcracking, consistent with observations of increasing ultrasonic velocity anisotropy in larger samples¹⁵⁻¹⁸. The difference from earlier studies is that we have independently verified the underlying mechanisms of compaction and dilatancy from the strain fields, hence validating hypotheses derived from these earlier studies.

Our combined direct (μ CT) and indirect (AE) *in-situ* observations show a tendency for the AE sources to be initially more broadly distributed throughout the sample (strain increments 1-19) and then to progressively localise along the three candidate shear zones close to peak stress and, eventually, the critically-oriented shear zone (strain increments 19 onwards). Although we were limited by the number of AE events (<200 events, representing ~5% of recorded AEs) we could locate with our location algorithm, this overall trend is consistent with previous *in-situ* observations of AE localisation in larger samples^{1,13,15-18}. Polarity estimates from our AE dataset were unreliable and insufficient, preventing us from distinguishing AE source types and testing hypotheses regarding the relationships between AE amplitude and source type or between AE source type and local strain magnitude. However, the strain field evolution (Figs. 3 and 4 and Supplementary Figs. 16 and 17) is broadly consistent with earlier observations of AE source types during shear failure^{6,17,18}, which show a high proportion of tensile-type events approaching peak stress, decreasing during failure in favour of an increasing proportion of shear- and collapse-type events. Furthermore, the observed co-existence of tensile, shear and collapse micro-mechanisms within the shear zone, along with tensile microcracks emerging from pore collapse, Hertzian grain contacts and shear sliding, reflects the relatively high proportion of mixed-mode AE sources previously detected during shear failure^{6,18}.

Lines 324-350:

While the DVC correlation window size (316 μm) of approximately one grain size (250-400 μm) averaged over processes occurring at sub-grain size, DVC estimates of the displacement of these windows were accurate to sub-voxel ($>7.91\mu\text{m}$) resolution^{32,33,34}. We therefore expect the local strain values to be accurate and representative of sub-grain-scale deformation, but unable to discriminate between sub-grain-size micro-mechanisms. One reason for the observed lack of correlation between large AE events and large strains may be a combined spatio-temporal resolution constraint, whereby instantaneous AE sources of different types – occurring very close together over a time-frame shorter than the inter-scan time (~ 85 s) – may cancel each other out. For example, this would suppress local strain estimates in areas where pore collapse also initiated tensile pore-emanating cracks, within the shear zone where radial dilation co-occurred with axial compaction, or where synthetic shear sliding co-occurred with antithetic shear sliding. This would lead to instantaneous AE events being correlated with smaller strains than perhaps they should have been. All of these could potentially alter the strain–AE amplitude mapping shown in Fig. 5.

Locations of dilation were strongly correlated with those of shear strain throughout the processes of localisation under the conditions examined here ($P_{\text{eff}} = 20$ MPa). This applied equally to the exploration of candidate shear zones and the ultimate development of the final shear zone. Although localised compaction along the critically-oriented shear zone did occur (Supplementary Fig. 17; panel 24) this was significantly lower in intensity prior to the coherent sliding phase. Thus, shear zone development was primarily enabled by localised dilation. This is consistent with independent observation of dilatant shear zones in post-failure μCT images of dry Vosges sandstone and post-failure microscopy images of saturated Berea sandstone^{35,36} after deformation under a constant strain rate at pressures within the brittle regime ($P_{\text{eff}} < 40$ MPa). However, it is at odds with observations of mainly compactant shear zone development during deformation of Flechtingen sandstone^{17,18} at P_{eff} of 40 MPa under a constant AE event rate. These differences may be explained by inferences that the transition to shear-enhanced compaction occurs at the transition from the brittle regime to the semi-brittle regime^{35,36} ($P_{\text{eff}} \sim 40$ MPa), and could indicate that effective pressure conditions have more influence on the micro-mechanics than differences in loading rate.

Lines 360-372:

However, en-echelon tensile cracks are the first of the damage micro-mechanisms to occur as the shear zone moves across the sample, with the more aseismic processes of further dilation, rotation and cataclasis following behind. This two-stage fault weakening process is independently consistent with the slip-weakening curve for total observed local slip in the developing shear zone (Fig. 6b; green): a short, steep reduction in shear stress is followed by a longer, shallower reduction. Our observations of two-stage weakening are consistent with observations of near-tip weakening followed by long-tailed weakening in biaxial stick-slip experiments³⁹. We note that en-echelon tensile microcracking was not observed along the shear zone until tomogram 22 (i.e., once dilation intensities in Fig. 3a reach into the purple values), indicating a critical amount of dilation is required for microcrack localisation along the shear zone. Therefore, the region of low amplitude strain that precedes higher amplitude strain as the shear zone grows across the sample (Fig. 3a and Supplementary Fig. 4a) is likely not a breakdown zone in the micro-mechanical sense.

Figure 1

Why have you selected this AE rate (1 AE/sec)? Elaborate a bit further on the selection of this threshold.

The AE rate of 1 AE/s was selected through extensive in-house testing prior to the experimental campaign in order to find the ideal rate for effective feedback control – i.e., one that kicked in early enough to prevent dynamic failure but not too early to avoid a sudden increase in strain rate, and could be maintained throughout failure given the relatively small number of AE events emitted by the small-scale sample compared with previous studies on larger-scale samples. We have added the following text to the methods section (lines 472-487): This AE event rate was established through extensive in-house testing, prior to the experimental campaign, to be the optimum event rate for effective feedback control given the small sample size required for high-resolution μCT imaging (10 mm diameter x 25 mm length) and relatively few recorded

AE events (~3500) compared with earlier AE feedback control experiments that used larger-scale samples. These earlier studies on granite^{1,13,15} and sandstone^{13,17,18} recorded a wide range of event numbers (4000-70000) in samples of 50-76 mm diameter x 120-190 mm length. The samples used were dry, which is likely to increase the number and amplitude of events compared with a water-saturated sample (our case), but were conducted at higher confining pressure (40-50 MPa compared with 20 MPa in our case), which tends to suppress the number and amplitude of microcracks. The wide range of recorded events in these studies did not depend on rock type but may be related to the event rate used in each case, and therefore the duration of quasi-static shear zone development. However, it is difficult to compare our chosen optimum event rate with these studies since they do not state the AE event rate used. However, our protocol ensured that the AE feedback system took control of sample loading early enough to prevent dynamic failure but late enough to avoid a sudden increase in strain rate, and could be maintained throughout failure. This enabled...

Figure 1 caption

Please change second (1b) to (1c). **Changed.**

How have you defined the yield point? How the time intervals (i-v) in (1a) correlate with strain intervals (a-g) in (1c)?

We have changed the labels a-g to numbers reflecting the order in which the tomograms appear in the time-series to avoid confusion with the figure labels. This has also been done in Figs. 2 and 3, in all the relevant figure captions, and in the main text. In addition, we have amended the Fig. 1 caption text to address these questions as follows: **(a)** Evolution of differential stress, σ , and AE event rate, \dot{N} , with time. The plotted AE event rate was calculated from all recorded events, binned into 10 s time intervals. Consistent with previous studies^{73,74}, we identified five stages of deformation: (i) initial compaction and then quasi-elastic behaviour up to the yield point, (ii) strain hardening approaching peak stress, σ_p , (iii) damage zone localisation and strain softening beyond σ_p , (iv) sample weakening due to shear zone development through the sample, and (v) shear sliding along a contiguous sub-planar fault. The transition from constant strain rate loading (10^{-5} s^{-1}) to constant AE event rate loading ($1 \pm 1 \text{ AE s}^{-1}$) occurred early in stage (ii) shortly after the sample yield point, which was defined by the point at which the stress-strain curve deviated from linearity and the AE event rate accelerated beyond the steady but low rate observed during the elastic region (linear portion of the stress-strain curve). **(b)** Photograph of the failed sample showing the localised shear damage zone. **(c)** Differential stress plotted against axial strain. Number labels in (a) and (c) refer to the μCT slice and strain increment labels in Figs. 2 and 3, with tomogram 16 acquired at the yield point (transition from stage i to ii), tomogram 19 acquired at peak stress (transition from stage ii to iii), tomogram 22 acquired as microcracks localised along the critically-oriented shear zone (transition from stage iii to iv), tomograms 25 and 28 acquired during shear zone development, tomogram 31 acquired at the onset of coherent sliding (transition from stage iv to v) and tomogram 34 acquired during coherent sliding. Young's modulus, $E = 19.369 \pm 0.028 \text{ GPa}$, was calculated over the range shown. AE activity began at 40% of peak stress, σ_p , with initial strain localisation evident in the strain increments from $0.7\sigma_p$ onwards, and sample yield following at $0.85\sigma_p$. The AE feedback control (1 AE s^{-1}) modulated the strain rate from $0.93\sigma_p$.

You can potentially add the (i-v) vertical lines to (1c).

We chose not include the vertical lines in Fig. 1c for clarity of the figure because the lines ii-iii, iii-iv and iv-v would lie almost on top of each other due to the vertical nature of the stress-strain curve after peak stress. However, we have added the selected tomogram numbers to Fig. 1a for comparison. We hope that this, and the new text above relating the stages with the image numbers in the Fig. 1 caption is sufficient to address your concerns in this regard.

Have you thresholded any of the AE recorded amplitudes or do you present all recorded amplitudes in (1a)?

Figure 1a shows the AE event rate. In this figure, we show all recorded events binned into 10s intervals (see text added to the caption above). Supplementary Figure 10a shows the amplitude for every recorded AE event. The full AE catalogue contains only events above an instantaneous amplitude threshold (trigger threshold) of 280 mV (after pre-amp gain of 70dB) that we set in the acquisition system to ensure we were recording only AE events above the ambient noise threshold in the experimental hutch. This is described in the methods (lines 562-568) and we have added some text for clarity: The pre-amplification gain and trigger threshold were set at 70 dB and 280 mV respectively, with the aim of controlling the AE event rate effectively. The gain was determined from a benchmark pencil lead-break test in the laboratory prior to visiting the synchrotron, and the trigger threshold was determined by the ambient noise in the experimental hutch on the I12-JEEP beamline. Only AE events with amplitudes above the trigger threshold were recorded by the acquisition system, and AE event rates were calculated from the number of events recorded in 10 s time intervals throughout the experiment.

Figure 2:

Add the resolution of the XRT images (scale) somewhere in the picture. Add the movement of the piston (upwards or downwards, using an arrow indicating the applied global force). Both added. In (2a) I can see the movement of the hanging wall, but, you can, possibly, expand laterally the white dashed line so your readers can see the vertical boundaries of your vertical cross section. We have done this.

In (2a) the white dashed line on the shear band covers the grain texture you want to visualise. I have 2 suggestions here: a) show a zone instead of the core (i.e., two offset lines, similar to what you show at (2a), but due to the small size of the image one cannot see any grain scale activity clearly). b) show the vertical displacement field of this cross section (see Charalampidou et al., 2014 (IJRMMS)) – in that case, I would use a time interval that is closer to your end-product (XRT image you visualise here); this can be either the displacement field calculated between 1st and last XRTs or any smaller time interval you have calculated and can be of interest. I'm pretty sure that the displacement field can also capture the grain crushing shown on the bottom left side of the current vertical cross section.

For Figure 2, we opted for your suggestion (a) and enlarged the image as much as possible to show the grain scale activity. We have amended the Fig. 2 caption to incorporate these changes, as follows: Fig. 2. Micro-scale damage evolution close to and following peak stress. (a) Reconstructed 2D μ CT slice (x,y-oriented, where x and y are perpendicular to each other and to the direction of loading, and x is across-strike and y is along-strike of the shear zone) showing the plane (orange line) of a re-slice of the original μ CT volume (x,z-oriented, where z is the direction of loading) where the shear zone initially localised. The corresponding across-strike (x,z-oriented) re-slice is shown between the pale orange arrows with the shear zone highlighted by dash-dot lines. Zoomed-in view of the portion of the x,z-oriented re-slice contained within the solid box is shown between the pale blue arrows, highlighting the narrow shear zone that formed after peak stress (between the dash-dot lines), and the region of damage that formed after yield but before peak stress (dotted circle). (b) Further zoomed-in view of the across-strike slices (x,z-oriented) for selected tomograms, labelled with the number by which they appeared in the time-series (see Fig. 1 for the locations of these scans in the stress-time and stress-strain evolution, and Fig. 3 for the local strain increments following each of these scans) showing shear zone emergence and development in region shown by dashed box in zoomed-in view in (a). (c) Even further zoomed-in slices (x,z-oriented) highlighting the variety of micro-mechanisms involved in shear zone formation: numbers i-iii correspond to the dashed boxes in (b; slice 16). When the whole time-series is viewed as an animation (Supplementary Movies 1 and 2), the micro-mechanisms illustrated by the annotations are apparent.

The vertical displacement fields in the strain increments approaching peak stress do capture greater displacement in the bottom left side of the cross-section and we include a new figure (Supplementary Fig. 15) showing median projections of the incremental displacement fields summed over strain increments 14-19 approaching peak stress, and summed over increments 20-36 after peak stress, along with the

corresponding volumetric and shear strain fields, to highlight the influence of this region in the localisation of the critically-oriented shear zone. We refer to this figure in the amended results section (lines 109 and 120). The caption for this new figure is as follows: **Supplementary Figure 15: Median projections along-strike of vertical displacement (δz), volumetric strain ($\epsilon_{volumetric}$) and shear strain (ϵ_{shear}), summed over strain increments (a) 14-19 approaching peak stress, and (b) 20-36 after peak stress.** This figure highlights the influence of the region of enhanced vertical compaction in the bottom left part of the sample in (a) leading to localised dilation (tensile microcracking) and shear strain just below that region. This weakening in the microstructure facilitated bulk left-lateral motion, leading to strain localisation along and subsequent development of the critically-oriented shear band, as seen in (b).

2a_{ii}). Moreover, please, elaborate further on the time-lapse histogram. How have you defined the partial volume region and the rock matrix region? What covers the in-between them space on the histogram?

We have removed the time-lapse histogram from Fig. 2 since the original inclusion of the time-lapse histogram was as part of a figure showing the different phases segmented via global thresholding, where the partial volume voxels were defined by the region of the histogram that evolved with time. There was no in-between space on the histogram – that was an artefact of the ellipses we used to highlight the different regions in the image. The bounding grey values around the partial volume region were defined by the first grey-values in either direction at which the number of pixels remained almost the same throughout the experiment. However, this was a preliminary attempt at segmentation on 2D slices and so we didn't consider it reliable enough for publication. We are in the process of testing a variety of segmentation methods to establish the best way to segment the data, which we will present in a subsequent publication. Here, we chose to show the grey-scale images themselves to emphasise the detail they contain. The time-lapse histogram is therefore redundant since this paper is not concerned with segmentation of the microstructure.

In (2a_{iii}), what do the different identified mechanisms tells us about the processes? Can you also comment on the role of local porosity that facilitates or halts some mechanisms? You have very good resolution XRT images for that purpose. I'm basically suggesting a couple of sentences that relate the observed mechanisms with the initial pore space (that evolves as well) with deformation.

Relating the observed mechanisms with the initial pore space in a quantitative way requires segmentation of the pore space. As discussed above, segmentation is out of the scope of this paper. Instead we provide a qualitative description of the role of local porosity facilitating/halting some mechanisms, based on observations of the grey-scale tomograms and evidence from comparing the vertical displacement fields with the shear strain and dilation fields. We have amended the results section as follows (lines 98-182):

Initially, diffuse elastic compaction was observed throughout the microstructure (Supplementary Figs. 6b and 17). AE activity preceded initial localisation of dilation and shear strain during early loading [stage (i) in Fig. 1; Fig. 3 panel 13], consistent with previous AE studies on porous rocks using much larger samples^{5,6,25} or *in-situ* μ CT imaging of smaller samples^{22,23}. Between yield and shortly after peak stress [stage (ii); Fig. 3 panels 16-19], the spatial distribution of shear strain closely followed that of dilation, and competing strain clusters localised along three distinct conjugate planes of similar amplitude and dip (30° to maximum principal stress; typical of optimally-oriented faults in nature) but variable strike, indicating self-organised exploration of candidate shear zones. These direct observations highlight the exploratory nature of emergent localisation in a complex system. Microcrack damage initiated towards the bottom end of the sample (Fig. 2a) where a region of localised compaction, evident from enhanced vertical displacement (Supplementary Fig. 15a), led to the collapse of some pores and facilitated the subsequent nucleation of pore-emanating micro-cracks. Some of these cracks traced grain boundaries (inter-granular cracks) while others intruded into whole grains (intra-granular and trans-granular cracks). Some trans-granular cracks initiated at loaded grain-grain contacts, most likely due to local Hertzian contact forces within larger scale force chains of accumulated stresses. These cracks formed subparallel to the loading axis, were no longer than two grain diameters, and were observed to cluster towards the bottom end of the sample (Fig. 2a), in the region of competing strain

clusters (Fig. 3; panels 16-19) approaching peak stress. The enhanced vertical displacement in this part of the microstructure facilitated bulk shear movement of the top part of the sample towards the weakened region of compacting porosity and tensile microcracking. This, in turn, led to further strain localisation along the candidate shear zone that was critically-oriented for failure, and coherent relative movement of the 'hanging wall' above the shear zone (Supplementary Fig. 15b).

In stage (iii), dilation and shear strain concentrated along the critically-oriented shear zone soon after peak stress (Fig. 3 panel 22), preceded by a brief hiatus in the dilation and shear strain rate (Supplementary Fig. 4; panel 20). This hiatus was consistent with a similar hiatus observed in AE event rate shortly before failure²⁶, and in our case was associated with a brief increase in the rate of diffuse compaction (Supplementary Figs. 16 and 17; panel 20). The critically-oriented shear damage zone emerged spontaneously from the self-organised localisation of numerous, narrow en-echelon tensile microcracks that nucleated simultaneously along the whole length of the emerging shear zone (Fig. 2b and c; slice 22) due to localised, high amplitude dilation and shear strain (pink and green regions respectively in Fig. 3; panel 22 and Supplementary Fig. 4; panels 21 and 22). These en-echelon microcracks were, individually, predominantly confined to single whole grains and originated from pores and Hertzian contacts. As tensile damage mechanisms localised increasingly on the shear zone, initial diffuse compaction throughout the sample was swamped by localised dilation and shearing on the shear zone (Supplementary Figs. 4, 6, 16 and 17; Supplementary Movies 3-8). This was marked by the emergence of a fat tail in the respective frequency-amplitude distributions, which eventually became bimodal (Supplementary Fig. 5).

In stage (iv), the shear zone developed along-strike (Figs. 3 and 4; Supplementary Fig. 12). It developed with a degree of curvature, consistent with shear zone development on a crescent-shaped front revealed by AE locations in granite²¹. This along-strike development, together with the observed variation in strain intensity within the shear zone, is contrary to the assumptions of the breakdown zone model, which assumes a uniform slip distribution across a sample-sized fault (see Methods). Dilation and shear strain were highly correlated in the shear zone (Fig. 3), consistent with several micro-mechanisms co-existing to accommodate bulk shear motion (Fig. 2c; slices 22-31). These included the nucleation of pore emanating and Hertzian en-echelon, tensile cracks along the shear zone, which facilitated further downslope bulk shear movement of the top part of the sample. The opening of these cracks caused some of the new tensile cracks to widen, producing dilation and new pore space, and promoted synthetic sliding parallel to the shear zone, on favourably-oriented cracks. This, in turn, led to the development of tensile wing cracks at the tips of these sliding shear cracks. These micro-mechanisms are consistent with previous experimental observations and existing microcrack nucleation models⁷⁻¹¹. Bulk shear motion along the failure plane also caused some en-echelon tensile cracks to rotate away from the shear zone orientation. These were cracks that had dilated sufficiently to allow neighbouring grain fragments to rotate with the bulk shear motion of the top part of the sample. Some of these grain fragments remained attached as asperities to the walls of the shear zone, while others broke away with continued bulk shear motion. Rotation prevented further tensile crack propagation in the axial direction, and supported the walls of the shear zone to maintain a finite thickness of up to one grain diameter throughout failure, although a few individual cracks extended up to two grain diameters. It also facilitated antithetic motion along cracks oriented conjugate to the principal shear zone, including some resembling Riedel shear zones. Some rotating fragments moved freely in the newly generated pore space, antithetically (conjugate to the principal slip direction) relative to their neighbouring grains but without contact between them. These movements are expected to be aseismic. Other fragments were close enough for their crack surfaces to remain in contact during antithetic sliding against each other. We expect these movements to be seismic. Local crack/grain rotation with associated antithetic motion occurred frequently along the length of the shear zone and was apparent in every vertical slice along the strike direction. The grey-scale μ CT images show that this mechanism was most prevalent within the shear zone on the side of the sample where microcrack localisation initiated along the emerging failure plane. Further along strike, parts of the shear zone became narrower (less than one grain diameter), with vertical displacement increasingly accommodated along more steeply dipping and narrow tensile and shear fractures via a 'wing-

crack' style mechanism. Rotation was still apparent in less steeply dipping regions, although there was an overall decrease in the number and size of rotating grains/cracks, and in the total angle rotated, further along strike. All the mechanisms just described led to grain fragmentation (cataclasis), and generated a proto-cataclasite within the shear zone as whole grains disintegrated, partial grains fractured off the shear zone walls, and fractured grains filled cavities (Fig. 2c; slices 25-34). Ongoing cataclasis resulted in compaction along the shear zone, spatially correlated with dilation and shear strain, but with smaller amplitude (Supplementary Figs. 4 and 17; Supplementary Movies 3-8). In addition to compaction of new pore space that had been generated by dilation within the shear zone, grain/crack rotation was a key facilitator of shear zone compaction. Increased rotation caused the crack dip angles to decrease, discouraging continued shear motion between grain fragments while encouraging compaction and closure of shallow dipping cracks during coherent sliding. These observations highlight the significant aseismic contribution (i.e., rotation of freely moving grain fragments, and other silent grain rearrangement) to the overall failure process, relative to seismic mechanisms (i.e., cracking and shear sliding⁶).

Figure 3:

How have you defined a) yield; b) final localisation (?) – maybe you need to find another term here; something like pre-stress drop or something similar; same for the other terms – is it shear zone propagation or better development? I think that the development starts from interval c; then, how are you sure that 'coherent sliding' is not taking place at the top part of the shear band also in what you call propagation interval (based on visual inspections of your incremental strain fields)? I believe that if you well define this terminology, then it is fine to go with it. You might also like to further talk about the observed micro processes that potentially coexist at different patches of your evolving deformation band during subsequent time intervals.

We have removed references to the terms final localisation and shear zone propagation throughout, and referred instead to 'localisation along the critically-oriented shear zone' and 'shear zone development'. We have also amended the labels in Fig. 3 to reflect the changes to the tomogram labels in Figs. 1 and 2 above and have added the following text to the figure caption: **Fig. 3. Selected 3D incremental strain fields from the onset of strain clustering (marked in Fig. 1c).** Incremental dilation, $\Delta\epsilon_d$ (blue-pink) and shear strain, $\Delta\epsilon_s$ (yellow-green) were calculated from digital correlation between successive pairs of μ CT volumes and are shown (a) parallel to strike (y,z orientation) and (b) perpendicular to strike (x,z orientation). The lower threshold of 0.0017 was set at four standard deviations from the mean of the error distribution of $\Delta\epsilon_s$ (Supplementary Fig. 6) and the upper threshold shows regions with strain >0.01 (maximum $\Delta\epsilon_s$ and $\Delta\epsilon_d$ were ~ 0.04 ; Supplementary Figs. 4 and 5). The thresholds were chosen to visually highlight regions of localised strain. Number labels correspond to those in Figs. 1 and 2, with the strain increment between the numbered tomogram and its subsequent neighbouring tomogram. Strain localisation began at $0.7\sigma_p$. The yield point here is the same as that shown in Fig. 1c, with the corresponding strain increment shown immediately following yield. The shear zone formed as crack localisation occurred along the critically-oriented plane in tomogram 22 (Fig. 2), and developed between strain increments 22 and 31, with patches of high intensity strain moving first up, then across and eventually down the sample, until high intensity dilation almost stopped. We defined coherent sliding as sliding along the whole shear zone, with the onset of coherent sliding at strain increment 31 since the intensity of dilation was significantly less in this increment than in previous increments, and this increment coincided with the point at which the rate of stress reduction slowed down in Fig. 1a. The full time-series of incremental dilation and shear strain from the onset of strain localisation is shown in Supplementary Fig. 4.

Lines 207-217:

Which Figure supports the indicated numbers for the fracture energy? We have added a reference to Fig. 6c (line 373).

It is a bit unclear to me whether in this paragraph you talk about the energy budget in general (so fracture, radiated and heat energies) or only the fracture energy. You might also need to further elaborate on the seismic and aseismic classification.

Here we are comparing only the bulk and local fracture energies and discussing the discrepancy between the two. We have amended the discussion text as follows (lines 374-394): Since G_{c-iii} for local slip in a propagating shear zone is only 50-68% of our two G_c estimates (i and ii respectively) for uniform slip on a sample-sized fault, it's possible that significant slip distributed throughout the rest of the sample, including the candidate shear zones, may account for the discrepancy between the local and bulk shear fracture energies (~32-50%). However, the average off-fault to on-fault incremental shear and volumetric strain ratios during failure are only 9% and 3% respectively (Supplementary Table 1); evidence for only a small amount of distributed slip, and consistent with ratios of off-fault dissipated energy to on-fault shear fracture energy in granite³⁷. This means that, although not all axial strain was accommodated by shear slip in the fault plane (contrary to one of the assumptions of the breakdown zone model – see Methods) and therefore the estimated bulk shear fracture energy for the eventual fault plane is an upper bound, slip in the shear zone still dominated the total. Therefore, the remaining 20-38% of additional bulk shear fracture energy must be accommodated by slip within the shear zone itself that does not occur by dilation or shear mechanisms. Such mechanisms are more likely to be aseismic (i.e., crack and grain rotation, and other silent grain rearrangements, including antithetic relative motion of non-touching grains and grain fragments), rather than seismic (i.e., dilation induced by tensile micro-cracking, and shear sliding along narrow, rough crack or grain boundary surfaces that are in contact with each other or have asperities between them). Although more detailed quantification of the contribution of rotation to the shear fracture energy is required, we show that slip due to rotation for individual cracks can be as large as $77 \pm 29\%$ of the local relative slip (Supplementary Table 2), and hence likely accounts for a significant proportion of the shear fracture energy.

If this paragraph is on energy budget in general, what kind/type of uncertainties you anticipate with regards to the radiated energy given that you have worked with only 2 sensors and given that you considered only the 5% of the recorded AE?

We restricted ourselves only to the fracture energy in this discussion rather than the energy budget in general. We didn't look at the radiated energy from the acoustic data. However, we would expect the radiated energy to be proportional to the total moment release. It might be interesting to compare the radiated energy from the amplitudes of the 5% of AE used for the location analysis with the radiated energy from all recorded AE, but this is out-with the scope of this paper and would form part of future work.

Lines 221- 222: Local crack rotation with antithetic slip (Fig 2c-iii) offers an additional mechanism for local stress rotation. How frequent this is within your sample? Rotation can be accommodated where the local dilation allows it. Is it something you observe frequently in the x-ray images?

Yes, this mechanism occurs frequently throughout the whole shear zone once localised, en-echelon tensile cracks have nucleated. We have added the following text to the results section (lines 162-170): Local crack/grain rotation with associated antithetic motion occurred frequently along the length of the shear zone and was apparent in every vertical slice along the strike direction. The grey-scale μ CT images show that this mechanism was most prevalent within the shear zone on the side of the sample where microcrack localisation initiated along the emerging failure plane. Further along strike, parts of the shear zone became narrower (less than one grain diameter), with vertical displacement increasingly accommodated along more steeply dipping and narrow tensile and shear fractures via a 'wing-crack' style mechanism. Rotation was still apparent in less steeply dipping regions, although there was an overall decrease in the number and size of rotating grains/cracks, and in the total angle rotated, further along strike.

Lines 242-245: This discussion highlights the potential for re-examination of the microstructures and inferred mechanisms associated with larger-scale seismic and aseismic processes considering the results presented here, in particular, the conclusion that seismic events miss important grain-scale mechanisms governed by

kinematics before and during shear failure. Do you believe that the occurring micro-processes would have been the same if you did not apply the AE feedback? How the AE feedback control relates to larger-scale seismic and aseismic processes?

We cannot make a direct comparison since we don't have observations of the dynamic process for reasons stated in the manuscript, apart from analysing the post-test scans in each case, where we can do a like for like comparison. Detailed comparison between a constant strain rate experiment and this constant AE event rate experiment is the subject of ongoing work. We have added the following text to the discussion (lines 396-406): The quasi-static nature of the shear zone development presented here suggests that our results may be most directly applicable to slow earthquakes, but it is also possible that all the observed processes also occur during dynamic failure, just much more rapidly and potentially all together. Fault propagation rates independently inferred from high resolution AE records¹⁵ were three orders of magnitude smaller during AE feedback loading than during constant strain rate loading (3-14 μs^{-1} and 1-18 mms^{-1} respectively). However, during constant strain rate loading fault propagation underwent a stable growth phase between initial fault localisation and unstable dynamic propagation, implying mechanistic similarities between the two loading rates during this phase. Unfortunately, direct μCT observations of the dynamic process at the temporal resolution required for comparison are not available to resolve this issue, for reasons stated in the introduction.

Line 269: Can you mention the gain of the pre-amplifiers used, also in the main text?

The gain is mentioned in the Methods section (lines 562-564): The pre-amplification gain and trigger threshold were set at 70 dB and 280 mV respectively, with the aim of detecting sufficient events above ambient noise to control the AE event rate effectively.

We have also amended the results section as follows (lines 212-213): We recorded ~ 3600 AE events above ambient noise (instantaneous amplitude threshold of 280 mV at 70 dB pre-amplification gain) using axially-located P-wave sensors (see Methods).

Line 329: Can you provide further details about the 2-channel monitoring system (model etc)? Can you also provide some further details on the frequency of the PZE and potentially the width of the crystal (or some references)? Please add to the main text that these are P-wave sensors.

We have amended the methods section as follows:

Lines 454-458: ...two piezoelectric P-wave transducers, positioned axially (Supplementary Fig. 1), to passively detect acoustic emissions (AE) and actively monitor ultrasonic velocities. These sensors were connected to a two-channel Applied Seismology Consulting Ltd (ASC) micro-seismic monitoring system by means of two ASC pre-amplifiers (full details given below in the description of acoustic emission recording and analysis).

Lines 555-562: Detected AE signals were received by the pair of 'Glaser-type' P-wave piezoelectric transducers⁶². These transducers are broadband, with a flat response spectrum between 500 kHz and 2 MHz, and conical in shape, with a contact area 5 mm in diameter. The detected signals (in Volts and proportional to the true normal displacement of the received elastic wave) were amplified and sent first to the ASC Trigger Hit Counter, which triggered recording of events based on an instantaneous amplitude threshold, and then to the ASC Cecchi Acquisition Unit, which recorded the full event waveforms (at 50 MHz, 12 bit acquisition with 128 kilo-samples per channel), including arrival time, amplitude and first-motion information.

Line 364: SPAM – does the software consider the grain rotation when calculating the strain fields? Or else, do you have any calculated rotation field?

SPAM calculates the full deformation matrix which includes the antisymmetric part (rotations) and the symmetric part (strain). There are existing functions to separate the strain fields from the rotation fields of the displacement tensor, but they only output the strain fields. Obtaining the rotation fields requires

additional scripting in python. We therefore did not output the rotation fields when originally running the DVC. We observed grain/crack rotation qualitatively in the grey-scale images only after the original strain fields were output, and as a result of the observations reported in this paper, scripting to obtain the full 3D rotation fields is the subject of ongoing work which will be reported in a subsequent publication.

Lines 385-387: A stress drop of 1 MPa; The latter stress drop is an intermediate stress drop encompassing 23 different studies of natural, mining induced, and fracking induced seismicity and laboratory AE events. How this stress drop number relates to your experiment? Why such a stress drop? Please provide some comments to link this with your experimental data.

In our analysis the amplitude measurements are relative. The recorded amplitude is proportional to normal displacement (McLaskey and Glaser, 2012). By fixing the stress drop and the source dimension, one can fix the seismic moment (line 603). Citing precedent in Dresen et al. (2020) we fixed the seismic moment of the largest amplitude AE event, and scaled all other AE events to it, based on their relative amplitudes. Specifically, we assumed a Madariaga source model of maximum crack radius observed in the tomograms ($2r = 800 \mu\text{m}$, equivalent to two grain diameters) and further assumed a stress drop of 1 MPa, an average value that is remarkably scale invariant over 19 orders of magnitude in seismic moment (Supplementary Fig. 14, with our data shown in the thick red box). By plotting the resultant seismic moment vs the corner frequency on this figure, we can see the implied stress drop range is 0.1 to 10 MPa, also similar to those in other published work (straight, dashed lines in Supplementary Figure 14) so we are confident our calibration captures the average behaviour and the data scatter reasonably.

We have amended the methods for clarification (lines 603-614): The estimated maximum scalar seismic moment was $1.5 \times 10^{-3} \text{ Nm}$, using the Madariaga source model⁶⁴ with a radius, r , equal to one-half the maximum observed crack size ($2r \sim 800 \mu\text{m}$) and a stress drop of 1 MPa. This stress drop lies in the middle of the range of stress drops seen in 23 different studies^{65,66} of natural, mining induced, and fracking induced seismicity and laboratory AE events, with data spanning over 19 orders of magnitude in seismic moment (Supplementary Figure 14). Our results are also consistent with published laboratory AE studies that use Glaser-type P-wave transducers, similar to ours (Supplementary Fig. 14, with our data in red in the thick red box, and those of Blanke⁶⁵ et al. contained in the thin orange box). The maximum amplitudes of the AE envelopes were scaled to the maximum scalar seismic moment to obtain the scalar seismic moment distribution. With this calibration, our events spanned a stress drop range of 0.1 to 10 MPa (straight, dashed lines in Supplementary Fig. 14) similar to those seen at all scales in the figure.

Lines 404-405:

Given that the DVC has revealed strains not in a single plane (local variability, off plane volumetric and shear strains) can you elaborate further one of the assumptions, i.e., all axial strain is accommodated by shear slip in the fault plane (beyond peak)?

We have added the following to the discussion (lines 381-385): This means that, although not all axial strain was accommodated by shear slip in the fault plane (contrary to one of the assumptions of the breakdown zone model – see Methods) and therefore the estimated bulk shear fracture energy for the eventual fault plane is an upper bound, slip in the shear zone still dominated the total.

Figure 4 (and equivalent text):

I have gathered here several parts of the text, where you discuss Figure 4.

Lines 116-118: These observations highlight the significant contribution of aseismic mechanisms (i.e., rotation) to the overall failure process, relative to seismic mechanisms (i.e., cracking, stick-slip sliding; 6).

Lines 139- 142: Many small events occurred in regions of high local strain (Fig. 4 dashed black ellipses), consistent with deformation being primarily aseismic in the shear zone, while many large events occurred in regions of low local strain in the bulk (Fig. 4 and Supplementary Figure 12).

Lines 166-167: For the AE location (definition of the unique position of the AE) you assume that each AE occurred at the largest local (shear and/or volumetric) strain within its circular hyperboloid.

Line 198: moderate/small events occurring in regions of large directly measured strain

Lines 202-203: therefore, the AE source amplitude is not necessarily representative of the local strain

Lines 354-357: Given the large correlation between locations of large volumetric strain vs. deviatoric strain, AE locations were defined within the same pixel whether constrained by volumetric or deviatoric strain for 85% of the located AE.

Lines 360-363: Whilst the assumption of linking AE to the largest local strain should bias AE locations to be linked to larger local strains, the results showed that, effectively via proof by contradiction, the largest AEs do not occur at locations of large local strain, with the locations of large local strains being linked to small/moderate AE (Fig. 4). This observation, in turn, is evidence that deformation is primarily aseismic.

My comments:

Fig 4 shows (for the 5% of the recorded AE events) that larger (higher amplitude) AE events occurred in regions of lower local shear and volumetric strains whereas smaller AE events occurred in regions of higher local shear or volumetric strains. The authors suggest that the latter is consistent with deformation being primarily aseismic in the shear zone.

In my opinion another explanation for the above observation can be the type of the AE events (or else their source mechanisms). Personal experience indicates that shear and tensile type events have usually lower amplitudes compare to compressive type events. Moreover, in saturated samples, signals are usually weaker, so you might have lost a big number of events also because of that. Going back to my source mechanisms argument, I'm expecting shear and tensile type events in regions that are characterised by high shear and high positive volumetric strain (i.e., dilatant patches). So, this means that lower amplitude AE events (shear and tensile) co-exist with high shear and high volumetric (dilatant strains) – which is what you have in Fig 4. Compressive events (pore collapse and/or grain crushing) have larger amplitudes and I expect them to coincide with lower shear strains and higher volumetric (compactant) strains. Usually, the volumetric strains are not well resolved (at least as well as the shear strains). Moreover, you might have a compressive event (grain crushing or pore collapse) that generates a larger amplitude AE, but it is so local that is not resolved within your DVC window (or representative volume = 40 pixels x 7-8 um= 320 um). Q1: Haven't you observed any compactant (negative according to your convention) strains within those regions included in Fig. 4? And if yes, how do they correlate with the higher amplitude events? Q2: Are all volumetric strains in Fig 4 dilatant or are you plotting the absolute values of the volumetric strains? Have you divided with the absolute volumetric max? I would plot separately compactant and dilatant volumetric strains (and I would also divide each group with the max compactant or max dilatant). Q3: Have you considered any uncertainties related to what is visualised by the DVC because of limitations of the method? Here, I refer to case of compressive events (so larger amplitudes) and low compactant strains (shear strains are expected to be low here anyway).

Finally, in terms of processes, aseismic deformation involves rotations and large enough dilation to have floating fragments that rotate without getting in touch with other grains and cements. Any contact of the rotating fragment with other grains will generate further deformation and thus, will radiate acoustic energy, so it will not be aseismic anymore. How many of the above cases have you identified within your x-ray CT images? Is this a common (see also % of aseismic share you suggest in the energy budget session) observation? So, what I am suggesting here is to explore further the above argument and if this does not hold for your datasets, to continue with your existing arguments of aseismic deformation.

The conclusion that deformation is primarily aseismic is directly linked to the seismic strain partition factor, which is a quantitative result based on summing the scalar seismic moments from all events and dividing that by the average cumulative observed strain (both shear and volumetric). If there was any deformation unaccounted for due to resolution limitations of the DVC window, it would mean that we had underestimated the observed strains forming the denominator in the seismic strain partition factor. Hence, if we could account for any additional unresolved deformation, then the deformation would be even more aseismic than our current estimates. On the other hand, the seismic contribution also has resolution limits, which, if accounted for, could increase the seismic strain partition factor. This is being dealt with in a separate piece of work (Mangriotis et al., in preparation).

Your point about undetectable seismicity (e.g., during rotation, or from events below the detection threshold) is a good one, especially given the high b-value and therefore the dominance of smaller events, and the lack of correlation between regions of high strain and high amplitude AE events. Our reasoning was that because the largest strains were not reflected in the seismicity, then seismicity alone was not diagnostic of total deformation. Nevertheless, where we had referred to Fig. 5 as evidence that deformation is primarily aseismic (lines 221-222, 597 and 953) we have changed this to **indicative of deformation being primarily aseismic**.

In terms of magnitude threshold, the lack of correlation between seismic amplitude and local strain would not change significantly if we included the remaining 95% of detected (but not located) events, because those events are smaller than those we were able to locate. Hence, we have added the following to the discussion for clarification (lines 311-313): **However, this does not explain the relative absence of large AE events in high strain regions (Fig. 5); a behaviour that would not change significantly if we included the unlocated 95% of events since those events were smaller than the located events.**

We see many instances of aseismic rotation (non-touching grains) and seismic rotation (touching grains) throughout the shear zone as it develops, but the relative proportion of these mechanisms is impossible to quantify accurately without segmenting the microstructure. We have added the following to the results to highlight the co-existence of the two mechanisms (lines 158-162): **Some rotating fragments moved freely in the newly generated pore space, antithetically (conjugate to the principal slip direction) relative to their neighbouring grains but without contact between them. We would expect these movements to be aseismic. Other fragments were close enough for their crack surfaces to remain in contact during antithetic sliding against each other. We would expect these movements to be seismic.**

Q1: We explored the locations in space and time of the ~20 largest AE events relative to the local strain fields to establish whether regions of compaction are likely to be associated with large AE sources or not. We found that the largest events occurred in two clusters – the first between 2000 and 3500 s (strain increments 14-22; around peak stress; exploration of candidate shear zones), and the second between 4500 and 6000 s (strain increments 29-26; towards the end of shear zone development, into coherent sliding). For the first cluster, the kinematic bowls in which most of the AE events occurred were located in the bottom half of the sample (Supplementary Movie 9) where there was an extremely small amount of compaction compared with the amount of dilation and shear strain (Supplementary Figs. 4 and 17; panels 14-22). One of the largest AE events occurred in conjunction with the damage rate hiatus observed in the dilation and shear strain fields (Supplementary Fig. 4; panel 20) that was also associated with a slight increase in compaction (Supplementary Fig. 16; panel 20). However, the kinematic bowl for this event was located in the top half of the sample (Supplementary Movie 9), again in a region of very low compaction but larger dilation (Supplementary Figs. 4 and 17; panel 20). It is therefore unlikely that events in this cluster were compactant. On the other hand, the second cluster is associated with events occurring along the shear zone itself, associated with increased compaction, but also significant dilation and shear strain (Supplementary Figs. 4 and 17). Thus, it is possible that larger amplitude AE events in the second cluster could have been associated with larger compactant strains. However, we cannot conclude definitively whether larger AE were linked to compaction (or whether smaller AE were linked to dilation/shear) since the kinematic constraints were not

adequate for this purpose. Your point on finding the relationship between AE event type and amplitudes is very interesting, and we are still investigating whether it is something we can estimate with this dataset in independent work. Determining the event type is challenging because we only have two vertical sensors, so we cannot invert for the moment tensors. We attempted polarity analysis with only two sensors (using the active seismic to verify polarity directions), but found that first breaks are unstable, and polarity estimates unreliable. Independently, the unstable first breaks motivated our using a cross-correlation approach to estimate relative time arrivals between the two sensors for AE location. Therefore, we cannot yet distinguish AE into different event types, though we hope to continue efforts to resolve this issue. As a result, we cannot yet verify the hypothesis that shear and tensile AE events were lower amplitude than compaction AE events, nor how AE source mechanisms were linked to local strains. We have added the following to the discussion (lines 293-303): Polarity estimates from our AE dataset were unreliable and insufficient, preventing us from distinguishing AE source types and testing hypotheses regarding the relationships between AE amplitude and source type or between AE source type and local strain magnitude. However, the strain field evolution (Figs. 3 and 4 and Supplementary Figs. 16 and 17) is broadly consistent with earlier observations of AE source types during shear failure^{6,17,18}, which show a high proportion of tensile-type events approaching peak stress, decreasing during failure in favour of an increasing proportion of shear- and collapse-type events. Furthermore, the observed co-existence of tensile, shear and collapse micro-mechanisms within the shear zone, along with tensile microcracks emerging from pore collapse, Hertzian grain contacts and shear sliding, reflects the relatively high proportion of mixed-mode AE sources previously detected during shear failure^{6,18}.

Q2: AE location constraints to local strains were applied independently for maximum deviatoric strain (by convention always positive) and for maximum absolute volumetric strain (by convention negative for compression and positive for dilation). The maximum absolute volumetric strain constraint meant we guided the AE to the largest strain in an absolute sense (compaction or dilation) within its kinematic constraints, without making any assumptions about the volumetric source mechanism. It happened that the maximum absolute volumetric strain in each AE kinematic bowl was a dilation (not unsurprising given the strain distributions in Supplementary Fig. 5). Thus, each strain value in Fig. 5 is divided by the maximum absolute volumetric strain within its respective AE kinematic bowl, which also happens to be a dilation. So, first we estimated the locations based on the absolute maximum strain, and then assessed the actual value of the strain at each location, finding that all polarities were positive (Supplementary Fig. 13).

Q3: We agree that there is a limit to the deformation that can be quantified locally from the DVC window, and have added the following caveat (lines 324-336): While the DVC correlation window size (316 μm) of approximately one grain size (250-400 μm) averaged over processes occurring at sub-grain size, DVC estimates of the displacement of these windows were accurate to sub-voxel ($>7.91\mu\text{m}$) resolution^{32,33,34}. We therefore expect the local strain values to be accurate and representative of sub-grain-scale deformation, but unable to discriminate between sub-grain-size micro-mechanisms. One reason for the observed lack of correlation between large AE events and large strains may be a combined spatio-temporal resolution constraint, whereby instantaneous AE sources of different types – occurring very close together over a time-frame shorter than the inter-scan time (~ 85 s) – may cancel each other out. For example, this would suppress local strain estimates in areas where pore collapse also initiated tensile pore-emanating cracks, within the shear zone where radial dilation co-occurred with axial compaction, or where synthetic shear sliding co-occurred with antithetic shear sliding. This would lead to instantaneous AE events being correlated with smaller strains than perhaps they should have been. All of these could potentially alter the strain–AE amplitude mapping shown in Fig. 5.

One important thing to note however, is the proof by contradiction implied in our analysis. We bias the largest AE to occur at the largest strain possible (an assumption), and from their kinematic behaviour (physics) the data show us that large AE cannot be correlated with large strain (absence of points in the upper hand corner of Fig. 5). If there is a strain misrepresentation due to resolution limits of the DVC (either spatial or temporal), the conclusion from Fig. 5 could be different, however we cannot predict in what way. What

we can say is that at the spatial and temporal resolution of the strain increments, dilation and compaction often occur in the same location, as a result of the pore-emanating crack mechanism or co-occurrence of radial dilation with axial compaction in the shear zone. It might be possible to use the axial and radial components of the strain fields as a prior for the AE locations rather than the first strain invariant (volumetric), but that would form part of future work.

In response to your comment that 'Usually, the volumetric strains are not well resolved (at least as well as the shear strains)' we have added the following to the caption for Supplementary Fig. 7 (lines 135-144): For a given noise in the displacement field, the contribution of numerical noise in the calculation of the first and second strain invariants is higher for volumetric strains (first invariant) than deviatoric strains (second invariant). However, our estimate of DVC noise here shows that the error distributions are comparable between the two. In fact the strain value at one standard deviation from the mean is slightly smaller for volumetric strains (± 0.0005) than for deviatoric strains (0.0009), although the volumetric error standard deviation is twice that of the deviatoric error. This indicates that the numerical noise is smaller than other sources of noise in the strain measurement, although the sources of such noise are not immediately obvious. It is also evidence that our observed large strains are meaningful and discernible above the noise (the largest observed strain values being an order of magnitude larger than the largest observed error values).

REVIEWERS' COMMENTS

Reviewer #2 (Remarks to the Author):

Dear authors,

I would like to thank you for the time you made to address my previous comments. I do believe that your arguments, answers/comments to my questions and the revised version turns the manuscript ready for publication. Great work! Well done!

Best wishes,
Elma Charalampidou